# Blooming under Mediterranean Climate: Estimating Cultivar-Specific Chill and Heat Requirements of Almond and Apple Trees Using a Statistical Approach

**Isabel Díez-Palet [1], Inmaculada Funes [1], Robert Savé [1], Carmen Biel [2], Felicidad de Herralde [1], Xavier Miarnau [3], Francisco Vargas [4], Glòria Àvila [5], Joaquim Carbó [5] and Xavier Aranda [1,\*]**

1   IRTA, Institute of Agrifood Research and Technology, Torre Marimon, E-08140 Caldes de Montbui, Barcelona, Spain; isabeldpalet@gmail.com (I.D.-P.); immaculada.funes@irta.cat (I.F.); robert.save@irta.cat (R.S.); felicidad.deherralde@irta.cat (F.d.H.)
2   IRTA, Cabrils, E-08348 Cabrils, Barcelona, Spain; carme.biel@irta.cat
3   IRTA, Fruitcentre, E-25003 Lleida, Spain; xavier.miarnau@irta.cat
4   IRTA, Mas de Bover, E-43120 Constantí, Tarragona, Spain; fvargas45@gmail.com
5   IRTA-Mas Badia Foundation, E-17134 La Tallada d'Empordà, Girona, Spain; gloria.avila@irta.cat (G.À.); joaquim.carbo@irta.cat (J.C.)
*   Correspondence: xavier.aranda@irta.cat

**Abstract:** Climate change, and specifically global temperature increase, is expected to alter plant phenology. Temperate deciduous fruit trees have cultivar-specific chill and heat requirements to break dormancy and bloom. In this study, we aimed to estimate chill and heat requirements (in chill portions, CP, and growing degree hours, GDH, respectively) of 25 almond (30–36 years) and 12 apple (14–26 years) cultivars grown under a Mediterranean climate. The set included early and late blooming genotypes. Long-term phenological and temperature records were analyzed by means of partial least squares (PLS) regression. The main difference between early and late genotypes was chill requirement, ranging from 8.40 CP of early genotypes to 55.41 CP of extra-late genotypes. However, as chill requirements are quite easily attained by all almond cultivars in this study, year-to-year variations in actual blooming dates for each genotype are governed by variability of mean forcing temperatures. In contrast, different chill and heat combinations resulted in similar mean blooming dates for the studied apple cultivars. Mean temperature in both chilling and forcing phases determined their blooming time in the location studied. Overlaps and gaps between both phases were obtained. Despite some limitations, the PLS analysis has proven to be a useful tool to define both chilling and forcing phases. Nevertheless, since the delineation of these phases determine the total amount of CP and GDH, further efforts are needed to investigate the transition of these phases.

**Keywords:** flowering date; dormancy; PLS; climate change; deciduous fruit trees; chillR

## 1. Introduction

Dormancy in temperate deciduous fruit trees is a phase of the ontogenetic development that allows trees to survive unfavorable conditions, like cold damage during winter [1], while helping to preserve nutrients assimilated over the preceding season [2] and delaying the reproductive processes to guarantee the reproduction of the individual [3].

Flowering time results from the sum of two traits: chill and heat requirements (CR and HR, respectively [4]). The main function of these traits is to avoid trees breaking dormancy during the cold

season [2]. Thus, the adaptation of a crop to a specific location is intrinsic to the fulfilment of these agro-climatic requirements [3], which are known to be cultivar-specific. The accumulation of winter chill and heat is related to an adequate flower bud development and budburst. If the CR is not reached, it will be manifested with irregular, delayed and reduced budburst with an uneven flowering [5] that can endanger fruit production. However, some researchers have proposed that in some species, the insufficient chill during the chilling period could be compensated by high temperatures during the forcing phase [6–8].

Recent global warming has advanced spring phenological events of many woody species, whereas other species have shown an opposite behavior, delaying these events [9–11]. Funes et al. [12] have shown for apple trees that changes in blooming dates, both advancement or delay, can result from a mean temperature increase resulting from a trade-off between a delay in CR completion and advancement in heat requirement fulfilment. Furthermore, warming during the chilling phase could impede the cultivation of some species in their present locations if their CR are not met anymore [2]. In the case of temperate fruit trees, because of their need to fulfil CR to ensure homogeneous flowering and fruit set [3], the intensification of global warming may compromise the ability of many growers to satisfactorily produce the same array of crops as in the past [13–15]. On the other hand, such increases in temperature will allow the cultivation of species, such as citrus, that until now have been restricted because of too-low temperatures during winter.

One of the most vulnerable areas to climate change on Earth is the Mediterranean, where a higher-than-average increase in the mean temperature is projected for the next decades (IPCC-AR5, [16]). Catalonia (Figure 1), in the Northeast of Spain, is subjected to a Mediterranean climate, and is susceptible to a mean annual temperature increase of about 1.4 °C by mid-century (1.3 °C for winter, [17]). Almond (*Prunus dulcis* (Mill.) D.A. Webb) and apple (*Malus domestica* (Borkh.) Borkh.) are two main crops of relevant economic and social value in Catalonia, with 43,123 ha and 10,849 ha, respectively. In Spain, they represent 677,328 ha and 28.256 ha, with a recent increase of about 100,000 ha for almonds in mild and cold areas [18]. Catalonia contributed 16,957 t of almonds in 2018 to Spain's 339,033 t [19] (or 4832 of 53,119 without shell), the world's third largest (about 10% of the world's production) after USA and Australia and supplied 47.2% of apple production in Spain [12]. Both are widely cultivated fruit crops and of relevant economic importance around the world [20].

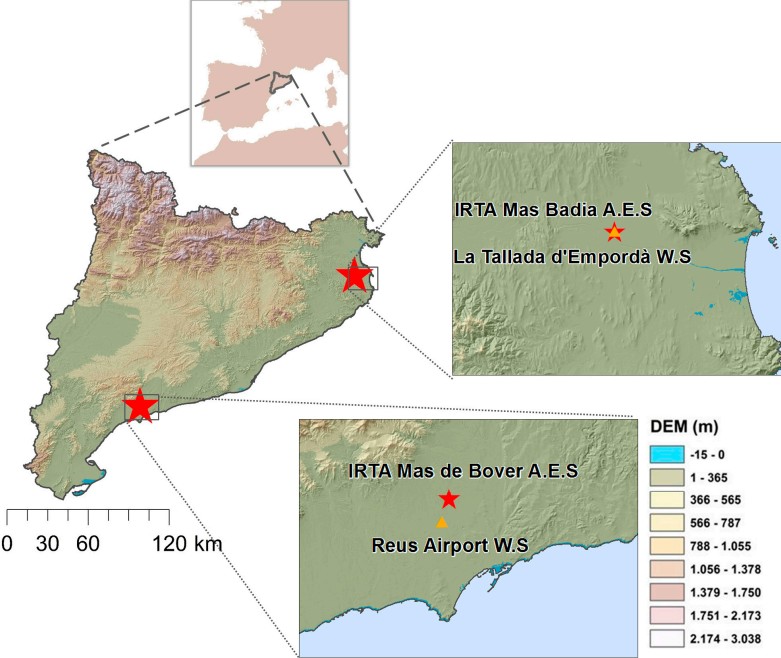

**Figure 1.** Location of the two IRTA (Institute of Agrifood Research and Technology) Agricultural

Experimental Stations (AES, red star) and the weather stations (WS, orange triangle) where flowering data and climate data were collected. Land orography is represented by using the Digital Elevation Model (DEM).

Several modelling approaches have been developed to quantify chill and heat effect on blooming and other phenological events in different fruit trees and climatic regions. The two most frequently used in the literature in warm regions, and specifically in Mediterranean climate areas, are the dynamic model [11,21] created for peach, and the growing degree hour (GDH) model [22]. Both attribute a specific parameter to quantify chill (chill portions, CPs) and heat effects (GDH). The combination of these two sub-models has been proven to perform well in studies with crops and locations similar to our case, although some odd results in the form of overlaps and gaps between chill and heat accumulation phases were obtained when they were not assumed to happen sequentially [2,12,23–27].

Since CR and HR values are useful to predict the adaptation of these crops, or their specific cultivars, to other locations, as well as predicting how climate change and increasing temperatures could influence their phenology, we estimated the chill and heat requirements of 25 almond and 12 apple cultivars including early and late blooming ones. Long series (>25 years) of blooming dates and temperature were used to estimate CR and HR by means of a statistical approach using partial least squares (PLS) regression [28], included in the chillR package developed for R (R Foundation for Statistical Computing, Vienna, Austria) [29].

## 2. Materials and Methods

### 2.1. Study Area

The study was carried out at two IRTA (Institute of Agrifood Research and Technology) Agricultural Experimental Stations (AES, Figure 1) located in Catalonia: IRTA-Mas de Bover (Constantí, Tarragona, Spain, 41°10′12′′ N, 1°10′10′′ E) and IRTA-Mas Badia (La Tallada d'Empordà, Girona, Spain, 42°3′9′′ N, 3°3′38′′ E). Both stations are subjected to the same Mediterranean coastal climate; hence, differences in monthly mean, maximum and minimum temperatures are small (Table S1 in supplemental material). Minimum and mean temperatures are somewhat lower in Mas Badia, although maximum temperatures are quite similar, and differences in annual values are hardly noticeable. A small difference has been recorded in annual mean precipitation: 530 mm in Mas de Bover versus 606 mm in Mas Badia.

### 2.2. Phenological Records

Almond and apple phenological data were obtained from Mas de Bover and Mas Badia AES, respectively.

Phenological flowering data of 25 almond cultivars from different geographical origins and different blooming times (i.e., early and late bloomers), were recorded for the 36-year period 1979–2015 by the IRTA Fruit Production Program in Mas de Bover. The date of the anthesis for 50% of the flowers (stage 65 in BBCH code) was used as the flowering date. Almond trees lacked a specific BBCH code until recently [30], and stages in Felipe et al. are generally used [31]. The F[50] stage was considered equivalent to BBCH 65. The range of the phenological observations in our dataset was continuous for all but 8 cultivars (Supplementary Table S2).

Bloom data of 12 apple cultivars were collected by the IRTA Fruit Production Program in Mas Badia from 1992 to 2018 (both years included). The apple bloom dataset from Mas Badia is the same dataset analyzed in Funes et al. [12] with some new records (up to five years) and cultivars added (Supplementary Table S3).

All phenological parameters were analyzed through an Analysis of Variance (ANOVA) using the ANOVA procedure of SAS software [32]. Means were separated where appropriate with a Duncan's multiple range test with a critical significance level of 0.05.

## 2.3. Climate Records

Meteorological stations with high-quality data coupled with phenological records were located nearby both AES (Figure 1). Daily mean, minimum and maximum temperature records from 1978 to 2015 were obtained from Reus Airport Weather Station from the Spanish Meteorology Agency (AEMET), located 2.8 km from IRTA Mas de Bover facilities. This dataset was used because it was longer that the dataset of the meteorological station of Mas de Bover AES Their equivalence was checked before use (correlation and average values). On the other hand, La Tallada d'Empordà Weather Station from the Meteorological Service of the Catalan Government (Meteocat) and located just in IRTA-Mas Badia facilities, provided same daily data for a 27-year period (1992–2018).

## 2.4. Temperature and Blooming Time Historical Trends

In order to analyze significant changes and detect trends in blooming dates of each species along both studied periods, the non-parametric Mann–Kendall trend test [33] was performed. This trend test is usually used in temporal series when the study parameters are slightly deviated from the normality [34]. The significance of the trends was evaluated at $p < 0.05$.

A Mann–Kendall trend analysis was also used to assess the significance of changes on yearly mean, minimum and maximum temperatures of the studied periods from both sites. The same procedure was done to analyze trends in the months when chilling (October–December) and forcing (January–March) phases occur. The significance levels used were $p < 0.001$ (***), $p < 0.01$ (**), $p < 0.05$ (*) and $p < 0.10$ (marginally significant, MS).

## 2.5. Estimating Chill and Heat Accumulation

Chill and heat accumulation were estimated by using the Dynamic Model [21,35] and the growing degree hour model [22], respectively. The combination of these two sub-models performed well in similar studies [2,12,23–27].

The dynamic model assumes that chill accumulates in a two-step process [21,35]. In the first step, low temperatures imply the formation of an intermediate thermolabile product that can be destroyed by high temperatures. Once a certain amount of such product has been accumulated, moderate temperatures contribute to its conversion to a permanent chill portion (CP). This model is well known to give good estimations of chill accumulation at different locations, and particularly, to work well in warm climates [3,23,36].

The growing degree hour model, which calculates growing degree hours (GDH), was the forcing sub-model. This model is based on the assumption that heat accumulates when hourly temperature is between a base temperature (4 °C) and a critical temperature (36 °C), producing the maximum heat accumulation at the optimum temperature (25 °C) [22,26].

## 2.6. Delineating Chilling and Forcing Phases: Cultivar-Specific Chill and Heat Requirements Estimation

To estimate chill and heat accumulation, we used the package 'chillR (version 0.62)' [2,37] in the statistical system R [29]. This package can use dynamic and GDH models to calculate chill and heat accumulation respectively, from an hourly series of temperature. However, chillR does not make any assumption about the relationship between both phases, and particularly, it does not assume a sequential model, which has been challenged in recent years [38]. The package uses partial least squares regression (PLS) to determine the dates, from a historical record, that had influenced a certain phenological event. It can also calculate chill portions and GDH, among other temperature-derived metrics. In the PLS part of chillR, the independent variables are the chill or heat accumulation in a specific day along all years in the record, and the dependent variable is the day of the year (DOY) a phenological event occurs (blooming date, in our case). A running mean of eleven days was used to smooth data. This is in contrast with the method used in Funes et al. [12], which assumed a sequential

model and used ordinary correlations to delineate the heat accumulation phase based on its mean temperature [39].

After the PLS regression, the recognition of the chilling and forcing periods was based on the two major outputs of the package: the variable importance in the projection (VIP) statistic, calculated for each independent variable to explain the changes in the dependent variables if this independent variable is omitted, and the standardized coefficients of the model [2]. A commonly used threshold of 0.8 for VIP [40] was applied to establish the onset and end of the chilling and forcing phases [2]. The standardized coefficients indicate the strength and the type of effect of chill or heat accumulated in a specific day on the flowering date [28]. For chilling and forcing periods, negative coefficients indicate that chill/heat accumulation results in an advancement of the blooming date. On the other hand, when these coefficients are positive, it indicates that the chill/heat accumulation on these days is delaying the blooming date [41]. Hence, average onset of the chilling and forcing phases was delimited as the first day of a period of consecutive days with persistent VIP values above 0.8 and negative standardized coefficients. However, these criteria were not met all along each phase for different statistical reasons, so the end of the phases was determined independently: the chilling and forcing phases were considered to extend until no more periods of persistent VIP values and negative standardized coefficients appeared, and the last day of such periods was taken as the average end of the respective phase. If the end of the forcing phase extended beyond the mean blooming date, which happened in most cases, we took the median date of all blooming dates recorded during the study period for each cultivar as the end of the forcing phase.

Once the average chilling and forcing phases were determined, the calculation of chill and heat accumulation from the initial date to the end date of each phase for each year was calculated using the chillR function 'chilling'. The mean, accompanied by its standard deviation, was taken as the estimation of the average CR and HR for every almond and apple cultivar.

### 2.7. Impacts of Chilling and Forcing Temperatures on Blooming Date

Blooming dates of apple and almond cultivars were plotted against mean temperatures for the chilling and forcing periods using the Kriging interpolation technique, frequently used in geostatistics. By interpolating these variables, we aimed to estimate values at locations (in this case, at bloom points in the plot surface) where no measurements were taken [41,42]. The color spectrum of the plots has to be interpreted as advancements or delays in the blooming time and isolines were created to represent homogeneous blooming dates. Closer points in the space tend to have more similar values than the distant ones.

## 3. Results

### 3.1. Temperature and Blooming Past Trends

Temperature trends were only significant for average and maximum, but not minimum. Annual temperature rose at a decadal rate of 0.28 °C and 0.5 °C for average and maximum temperature, respectively (Supplementary Table S4, Figure S1). No significant trends were observed in the phenological traits for any species, indicating that recorded blooming dates were independently distributed over the study period (Supplementary Tables S2 and S3).

Blooming data (Figure 2) show that, although the limits between the different categories are not very clear, the almond cultivars analyzed in this study follow the same pattern of flowering as those classified by Vargas and Romero [43] after adding 22 years of observations to their original 14 years of data. Significant differences were found between all blooming groups except between late and extra-late bloomers, which presented no statistically significant difference between their average blooming dates (Supplementary Table S5). This is consistent with an analysis of the blooming date variability among cultivars along the recorded years (Supplementary Table S6), which shows quite a compact group among late bloomers, and more differences among early and very early bloomers.

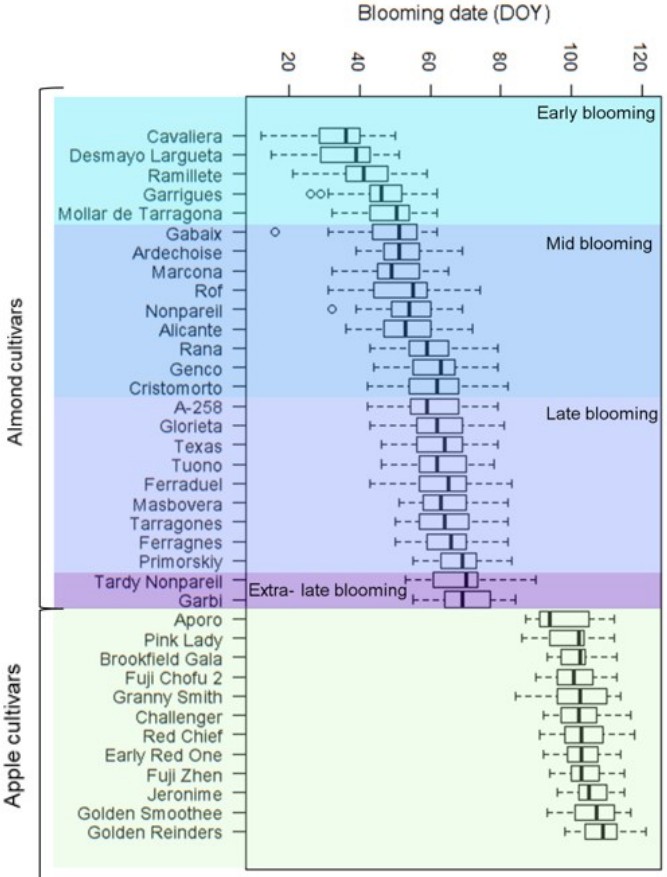

**Figure 2.** Statistical distribution of flowering date records (day of the year, DOY) of 25 almond cultivars cultivated in Mas de Bover from 1979 to 2015 and 12 apple cultivars cultivated in Mas Badia between 1992 and 2018. The left and the right part of each box are the first and third quartiles respectively, and the band inside the box is always the median. The ends of the whiskers represent standard deviation above and below the mean of the data. Small circles represent outliers. Almond cultivars are organized in early (turquoise), mid (blue), late (violet) and extra-late (purple) blooming time following the classification of Vargas and Romero [43]. Apple cultivars are represented in green.

Differences among average blooming dates of the apple cultivars were not so obvious (Supplementary Table S7), with clear differences only between the extremes, with Golden Reinders, Golden Smoothee and Jeromine on one side and Pink Lady and Aporo on the other. Clear, significant differences were found between the average blooming dates of almond and apple species (Supplementary Table S8), with all apple cultivars always blooming later than any almond cultivar.

*3.2. Chilling and Forcing Phases Delineation*

3.2.1. Almond Cultivars

Generally, all almond cultivars started their chilling phase in late October, specifically, on the 23–24 October, with the exception of 'Cavaliera', that started this phase a few days before (Table 1). Hence, from those days, all cultivars began to accumulate chill consistently and persistently. We discarded the small period of stable negative coefficients by the end of September–beginning of October as there was almost no chill accumulated in that period. As this does not depend on cultivar but on climate, we decided to discard this period in all cases. Moreover, including this period would only minimally affect CR as the chill accumulation in these days is very small.

**Table 1.** (**a**) Almond chill and heat requirements (CR, HR). Chill–Force overlap: % CR, percentage of chill accumulated when overlap with forcing phase started; % period, percentage of total chilling + forcing period in which chilling and forcing phase overlap; and % Chill–Force gap, percentage of total chilling + forcing period in which no chill nor heat is accumulated (gap). Chill and heat requirements (mean ± standard deviation). Chill requirements are expressed in chill portions (CP) and heat requirements in growing degree hours (GDH). CV: coefficient of variation associated to CR and HR. Cultivars are organized according to their average flowering date (from earliest to latest). (-) in column % Chill–Force overlap means that 100% of the chill was accumulated before the start of the forcing phase. (**b**) Apple start and end day of the chilling and forcing phases, % of accumulated chill when forcing phase started, % of the period with overlap (Chill–Force overlap) and % of the period were gap between phases occurs (% Chill–Force gap). Chill and heat requirements (mean ± standard deviation). Chill requirements are expressed in chill portions (CP) and heat requirements in growing degree hours (GDH). CV: coefficient of variation associated to CR and HR. Cultivars are organized according to their average flowering date (from earliest to latest). (-) mean that 100% of the chill was accumulated when the forcing phase started.

(**a**)

| Cultivar | Chilling Phase | | | Forcing Phase | | | Chill-Force Overlap | | % Chill-Force Gap | CR (CP) | HR (GDH) |
|---|---|---|---|---|---|---|---|---|---|---|---|
| | Start | End | N° days | Start | End | N° days | % CR | % period | | | |
| Cavaliera | 20 Oct | 28 Nov | 40 | 24 Nov | 5 Feb | 74 | 81% | 5% | | 11.56 ± 4.03 | 7452 ± 1601 |
| Desmayo Largueta | 24 Oct | 24 Nov | 32 | 22 Nov | 8 Feb | 79 | 88% | 3% | | 8.40 ± 3.77 | 8552 ± 1741 |
| Ramillete | 24 Oct | 15 Dec | 53 | 4 Dec | 10 Feb | 69 | 68% | 11% | | 20.71 ± 4.72 | 6998 ± 1540 |
| Garrigues | 24 Oct | 17 Dec | 55 | 30 Nov | 15 Feb | 78 | 53% | 16% | | 21.91 ± 4.66 | 8054 ± 1811 |
| Gabaix | 24 Oct | 3 Dec | 41 | 15 Dec | 20 Feb | 68 | - | - | 10% | 13.35 ± 4.35 | 6824 ± 1421 |
| Mollar de Tarragona | 24 Oct | 14 Dec | 52 | 16 Dec | 20 Feb | 67 | - | - | 1% | 20.04 ± 4.68 | 6718 ± 1378 |
| Marcona | 23 Oct | 17 Dec | 56 | 17 Dec | 18 Feb | 63 | - | - | | 21.96 ± 4.66 | 6378 ± 1341 |
| Ardechoise | 24 Oct | 15 Dec | 53 | 10 Dec | 20 Feb | 73 | 85% | 4% | | 21.82 ± 4.50 | 6994 ± 1546 |
| Rof | 23 Oct | 15 Dec | 54 | 18 Dec | 24 Feb | 69 | - | - | 2% | 20.76 ± 4.72 | 6965 ± 1355 |
| Alicante | 24 Oct | 15 Dec | 53 | 16 Dec | 22 Feb | 69 | - | - | | 20.71 ± 4.72 | 6940 ± 1400 |
| Nonpareil | 24 Oct | 15 Dec | 53 | 16 Dec | 20 Feb | 67 | - | - | | 20.71 ± 4.72 | 7062 ± 1399 |
| Rana | 24 Oct | 14 Dec | 52 | 28 Dec | 28 Feb | 63 | - | - | 11% | 20.04 ± 4.68 | 6518 ± 1292 |
| A-258 | 23 Oct | 9 Dec | 48 | 7 Dec | 28 Feb | 84 | 93% | 2% | | 17.05 ± 4.52 | 8725 ± 1712 |
| Cristomorto | 24 Oct | 15 Dec | 53 | 16 Dec | 3 Mar | 78 | - | - | | 20.71 ± 4.72 | 8236 ± 1482 |
| Genco | 24 Oct | 27 Dec | 64 | 9 Jan | 4 Mar | 55 | - | - | 10% | 28.68 ± 4.90 | 5971 ± 1189 |
| Glorieta | 24 Oct | 31 Jan | 100 | 6 Jan | 28 Feb | 54 | 68% | 20% | | 51.58 ± 5.90 | 5654 ± 1177 |
| Texas | 24 Oct | 31 Jan | 100 | 7 Jan | 5 Mar | 58 | 69% | 19% | | 51.58 ± 5.90 | 6280 ± 1225 |
| Ferraduel | 24 Oct | 2 Feb | 102 | 29 Dec | 6 Mar | 68 | 56% | 27% | | 52.85 ± 5.95 | 7285 ± 1362 |
| Masbovera | 24 Oct | 27 Dec | 64 | 6 Jan | 4 Mar | 58 | - | - | 8% | 28.61 ± 4.90 | 6232 ± 1221 |
| Tuono | 24 Oct | 2 Feb | 102 | 29 Dec | 3 Mar | 65 | 56% | 28% | | 52.85 ± 5.95 | 6870 ± 1319 |
| Ferragnes | 24 Oct | 15 Dec | 53 | 16 Dec | 7 Mar | 82 | - | - | | 20.71 ± 4.72 | 8696 ± 1543 |
| Tarragones | 24 Oct | 31 Jan | 100 | 6 Jan | 5 Mar | 59 | 68% | 20% | | 51.58 ± 5.90 | 6370 ± 1238 |
| Tardy Nonpareil | 24 Oct | 6 Feb | 106 | 15 Dec | 11 Mar | 87 | 37% | 39% | | 55.41 ± 5.91 | 9444 ± 1658 |
| Primorskiy | 24 Oct | 2 Feb | 102 | 7 Dec | 10 Mar | 94 | 30% | 42% | | 52.85 ± 5.95 | 10,201 ± 1834 |
| Garbí | 24 Oct | 2 Feb | 102 | 7 Jan | 10 Mar | 94 | 68% | 20% | | 52.85 ± 5.95 | 7040 ± 1312 |
| | | | | | | | | | CV | 0.53 | 0.15 |

**Table 1.** *Cont.*

(**b**)

| Cultivar | Chilling Phase | | | Forcing Phase | | | Chill-Force Overlap | | % Chill-Force Gap | CR (CP) | HR (GDH) |
|---|---|---|---|---|---|---|---|---|---|---|---|
| | Start | End | N° days | Start | End | N° days | % CR | % period | | | |
| Aporo | 16 Nov | 12 Jan | 58 | 4 Jan | 4 Apr | 91 | 86% | 6% | | 37.79 ± 2.72 | 9232 ± 1557 |
| Pink Lady | 21 Nov | 16 Jan | 57 | 11 Feb | 12 Apr | 61 | - | - | 18% | 37.81 ± 2.25 | 8065 ± 1332 |
| Brookfield Gala | 8 Nov | 16 Jan | 70 | 24 Jan | 13 Apr | 80 | - | - | 6% | 44.48 ± 3.38 | 9501 ± 1556 |
| Fuji Chofu 2 | 11 Nov | 27 Jan | 78 | 16 Feb | 11 Apr | 55 | - | - | 13% | 50.26 ± 3.05 | 7471 ± 1191 |
| Granny Smith | 10 Nov | 19 Jan | 71 | 3 Feb | 13 Apr | 70 | - | - | 9% | 45.57 ± 3.15 | 8813 ± 1455 |
| Challenger | 11 Nov | 16 Jan | 67 | 9 Feb | 12 Apr | 63 | - | - | 15% | 42.98 ± 2.90 | 8202 ± 1347 |
| Early Red One | 10 Nov | 16 Jan | 68 | 26 Jan | 13 Apr | 78 | - | - | 6% | 43.58 ± 3.13 | 9349 ± 1530 |

**Table 1.** *Cont.*

(**b**)

| Cultivar | Chilling Phase | | | Forcing Phase | | | Chill-Force Overlap | | % Chill-Force Gap | CR (CP) | HR (GDH) |
|---|---|---|---|---|---|---|---|---|---|---|---|
| | Start | End | N° days | Start | End | N° days | % CR | % period | | | |
| Red Chief | 11 Nov | 12 Jan | 63 | 17 Feb | 13 Apr | 56 | - | - | 23% | 40.24 ± 3.01 | 7767 ± 1175 |
| Fuji Zhen | 11 Nov | 2 Feb | 84 | 12 Feb | 13 Apr | 61 | - | - | 6% | 54.44 ± 3.04 | 8190 ± 1319 |
| Jeromine | 11 Nov | 25 Jan | 75 | 15 Feb | 15 Apr | 60 | - | - | 13% | 42.98 ± 2.90 | 8349 ± 1263 |
| Golden Smoothee | 10 Nov | 15 Jan | 67 | 2 Feb | 17 Apr | 75 | - | - | 11% | 42.81 ± 3.21 | 9690 ± 1530 |
| Golden Reinders | 11 Nov | 15 Jan | 66 | 14 Feb | 19 Apr | 65 | - | - | 18% | 42.38 ± 3.06 | 9239 ± 1315 |
| | | | | | | | | CV | | 0.11 | 0.09 |

The end of this phase was diffuse especially in the late-blooming 'Glorieta', 'Texas', 'Ferraduel', 'Tuono', 'Tarragonès', 'Primorskiy', and the very late-blooming 'Garbí' (Figure 3, Supplementary Figures S2–S24). The estimated end date ranged from 24 November for the very-early blooming 'Desmayo Largueta' (Figure 3) until 6 February for the late-flowering 'Tardy Nonpareil' (Table 1), showing a large difference between different blooming time cultivars. The forcing phase started and finished earlier for the early-blooming cultivars than for the late-blooming ones. As the date derived from PLS analysis exceeded the median flowering date in most cases, the end of this phase was taken as the median flowering date, ranging from 5 February for the very-early blooming 'Cavaliera' until 11 March for the late-blooming 'Tardy Nonpareil' (Table 1). It is important to note that while the heat accumulation phase was clearly delineated, with negative coefficients and high VIP values all along the period, the chilling phase was less clear, with many periods between the beginning and the end of the phase presenting low VIP values or even positive model coefficients.

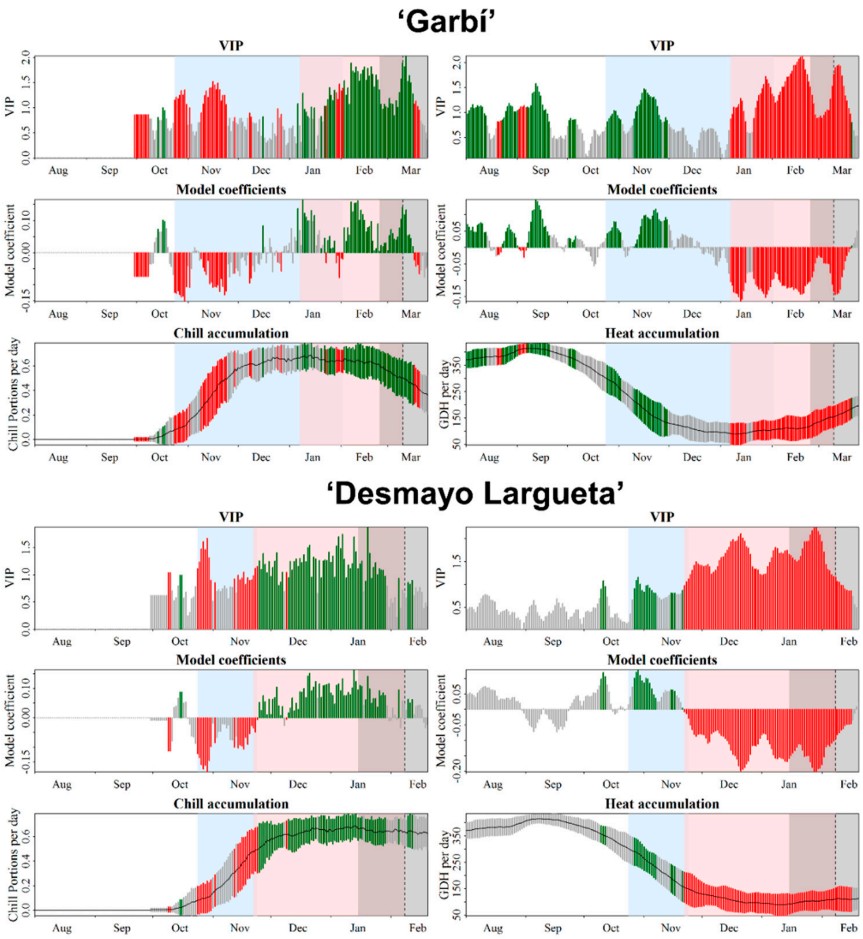

**Figure 3.** Results obtained from the partial least squares (PLS) regression analysis between blooming

dates and daily mean chill and heat accumulation for 'Garbí' and 'Desmayo Largueta' in Mas de Bover using the dynamic model and the GDH model. Chilling phase delineation on the left, forcing phase on the right. Top: VIP values (variable importance of the projection). Middle: standardized coefficients of the PLS model. Bottom: daily mean chill/heat accumulation in chill portions and growing degree hour (GDH) units, left and right respectively: the length of the bars in this panel indicates the standard deviation of the daily chill/heat accumulation. For all panels, red and green bars indicate VIP ≥ 0.8; red bars, the standardized coefficients of the model are negative indicating that chill or heat accumulated on that day (left and right panel, respectively) result in an advancement of the flowering date; for green bars, coefficients are positive an indicate flowering date delay. Blue and pink background colors emphasize the delineated chilling and forcing phases, grey background represents the period when flowering occurs along the years studied, and the dotted line marks the median date of all the blooming dates recorded from 1979 to 2015.

Two important findings were obtained while delineating the chill and forcing phases. The first one is the identification of an overlap period of the two studied phases in more than half of the cultivars: 'Garbí', 'Desmayo Largueta', 'Cavaliera', 'Ramillete', 'Garrigues', 'Ardechoise', 'A-258', 'Glorieta', 'Texas', 'Ferraduel', 'Tuono', 'Tarragonès', 'Tardy Nonpareil', and 'Primorskiy' (Table 1, Figure 3 and Supplementary Figures S2–S24). For those cultivars, the start of the forcing phase generally began when 50% or more of the chill had been accumulated, with the exception of 'Tardy Nonpareil' and 'Primorskiy' that started to accumulate heat when only about one third of the chill requirement was fulfilled (Table 1). Besides, for most of these cultivars, this overlapping period occupied 20% or more of the period between the start of the chilling phase and the end of the forcing phase. In the late and extra-late blooming cultivars, 'Tardy Nonpareil' and 'Primorskiy', the overlap supposed 39% and 42% of the total phenological period (Table 1). This behavior was more pronounced in the late-blooming cultivars than in the early and medium blooming ones.

The second finding is a gap period between phases in 'Gabaix', 'Mollar de Tarragona', 'Rof', 'Rana', 'Genco', and 'Masbovera' (Table 1 and Supplementary Figures S2–S24). This gap period was observed to last between 1% and 11% of the total period depending on the cultivar. Although no close relationship was found between the timing of bloom and the appearance of this gap, it was more frequent in the mid-blooming cultivars.

### 3.2.2. Apple Cultivars

Using the same criteria as for almonds, almost all apple cultivars started to accumulate chill continuously from 10–11 November, with the exception of 'Pink Lady', which started its chilling phase 10 days later (Table 1). After two months, in the middle of January, all cultivars stopped to accumulate chill except 'Jeromine' (Figure 4), 'Fuji Chofu2' and 'Fuji Zhen', which prolonged their chilling phase until late January or early February. 'Pink Lady' showed the shortest chilling phase and 'Fuji Chofu' and 'Fuji Zhen' the longest ones.

Establishing the chilling phase for this species was easier than for almond as continuous high VIP values and negative coefficients were persistent along the phase. However, the end of the chilling phase for 'Jeromine' and 'Golden Smoothee' was quite ambiguous since a period with low VIP values and positive coefficients at the end of the phase made it difficult to establish the end (Figure 4 and Supplementary Figure S33).

The beginning of the forcing phase differed among all cultivars and ranged between early January (4 January for 'Aporo', Figure 4) until mid–late February (17 February for 'Red Chief'). By contrast, the end of this phase was similar for all cultivars, by mid-April, with two exceptions: 'Aporo' and 'Golden Reinders', that ended their forcing phase on 4 April and 19 April, respectively (Figure 4 and Table 1). The limits of the forcing phase were more variable than their equivalents of the chilling phase. Another thing to consider is that the overlapping period observed in some almond cultivars is only observed in the cultivar 'Aporo', lasting only 6% of the dormancy period (Figure 4). The remaining

eleven apple cultivars showed a gap between chill and forcing phases that lasted approximately 5% of the total period (Table 1 and Figure 4). See figures for all cultivars in the supplemental material (Figures S25–S34).

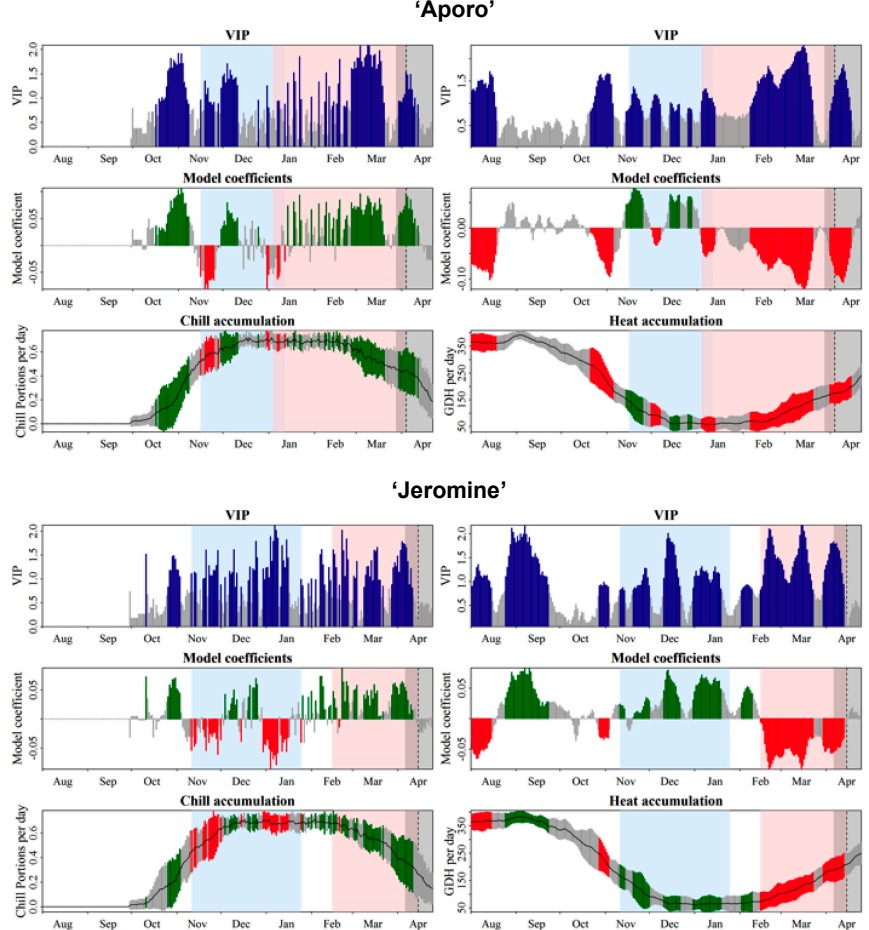

**Figure 4.** Results obtained from the PLS regression analysis between blooming dates and daily mean chill and heat accumulation for 'Aporo' and 'Jeromine' in Mas Badia using the dynamic model and the GDH model. See the caption in Figure 3 for the full explanation, except for the dotted line, which marks the median date of all the blooming dates recorded from 1992 to 2018

### 3.3. Chill and Heat Requirements

The amount of chill and heat accumulated during the chilling and forcing periods previously delineated were assumed as an estimation of the cultivar-specific CR and HR for every almond and apple cultivar.

#### 3.3.1. Almond Cultivars

CR and HR differed among all 25 almond cultivars (Table 1). In general, some cultivars had high CR together with elevated HR ('Tardy Nonpareil' and 'Primorskiy'), others, like 'Cavaliera' and 'Desmayo Largueta', showed low CR but elevated HR and vice versa for 'Glorieta', 'Texas' and 'Tuono' cultivars.

Significant differences in the CR between cultivar groups were observed (Supplementary Table S9) with early and mid-blooming cultivars differing significantly from late and extra-late varieties, with no other significant differences. The large variability in CR resulted in late blooming cultivars such as 'Tardy Nonpareil' extending their chilling phase by up to two months more than 'Cavaliera' (Table 1). Although no differences were found in HR between blooming groups (Supplementary Table S10),

high variability was also observed. In general, blooming date seems to be related to CR but not to HR in this cultivar collection. Hence, with some exceptions (e.g., 'Ferragnes'), the later the bloom of the cultivars, the higher their CR.

If cultivars are ordered according to their specific CR and HR, the formation of two distinct groups can be observed (Figure 5). Although these groups have similar HR, the variability on the specific CR of each almond cultivar marks the difference between these groups. This can be seen in Figure 6, which shows a statistically significant correlation between blooming date and length of chill accumulation phase across almond cultivars, and also with CR. However, no correlation could be found with length of heat accumulation phase or HR.

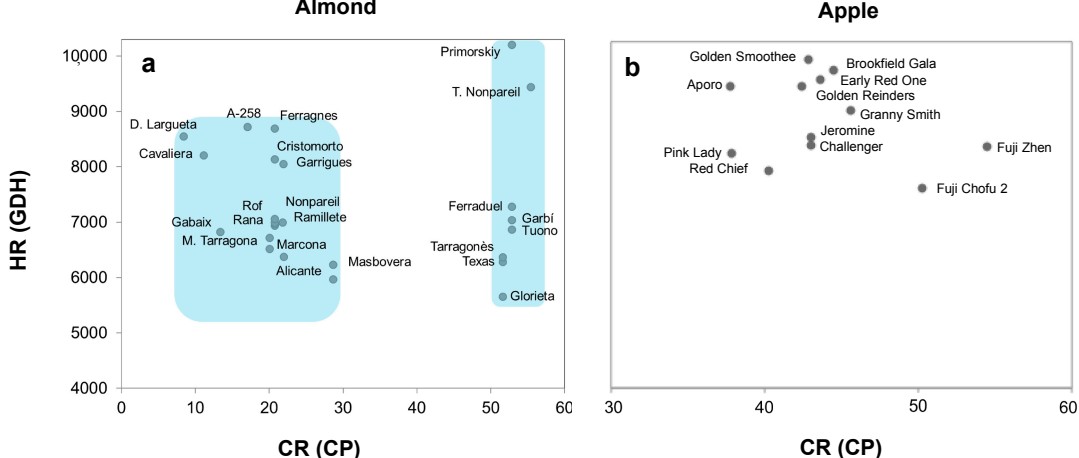

**Figure 5.** Almond and apple cultivars distribution according to their mean chill (CR) and heat (HR) requirements. Requirements calculated for (**a**) almond cultivars for the period 1979–2015 in Mas de Bover and (**b**) for apple cultivars for 1992–2015 in Mas Badia. GDH and CP stand for growing degree hour and chill portions, respectively.

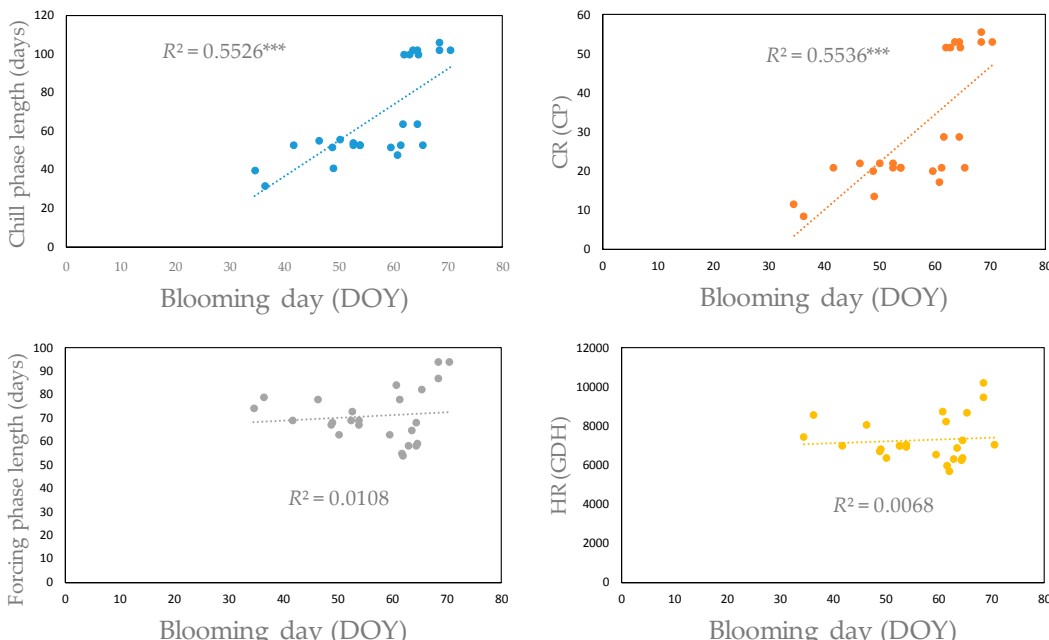

**Figure 6.** Correlations between almond cultivar blooming day and chill phase length (up, left), CR (up, right), forcing phase length (bottom, left) and HR length Almond and apple cultivars. DOY, day of year. *** Statistically significant $p < 0.001$.

### 3.3.2. Apple Cultivars

The variability of CR and HR was much lower in apple cultivars compared with almond cultivars (Table 1), and we could not observe a consistent pattern in CR and HR among cultivars: varieties like 'Fuji Chofu2' and 'Fuji Zhen' ranked higher in CR than in HR, whereas others like 'Golden Reinders' did the opposite. The lowest CR and HR were observed in 'Pink Lady' and 'Red Chief', which showed the earliest blooming dates. Likewise, 'Brookfield Gala', which presented one of the latest blooming dates, required a great amount of both chill and heat (Table 1).

Unlike almond cultivars, no groups could be established among apple cultivars according to their CR and HR, and therefore, we cannot determine if the differences in their blooming time depends more on their CR than on their HR. In fact, apple cultivars ranked between the two almond groups in CR, and in the upper part in HR (Figure 5). However, it seems that the blooming-advancing effects of the accumulation of chill and/or heat depends on the cultivar, by this we mean that to reach similar bloom dates, these cultivars presented different combinations of CR and HR (Table 1).

### 3.4. Impacts of Chilling and Forcing Temperatures on Blooming Dates

#### 3.4.1. Almond Cultivars

The influence of the interannual temperature variation on interannual blooming date variability inside a given cultivar can be analyzed in Figure 7. These three-dimensional plots present a surface interpolated from the actual blooming dates in relation to the temperatures recorded during the chilling and forcing periods previously identified by PLS regression (see the materials and methods section for more details). Isolines connect homogeneous blooming dates in this space, and the color spectrum has to be interpreted as advancements or delays in the blooming time as one moves from one isoline to another.

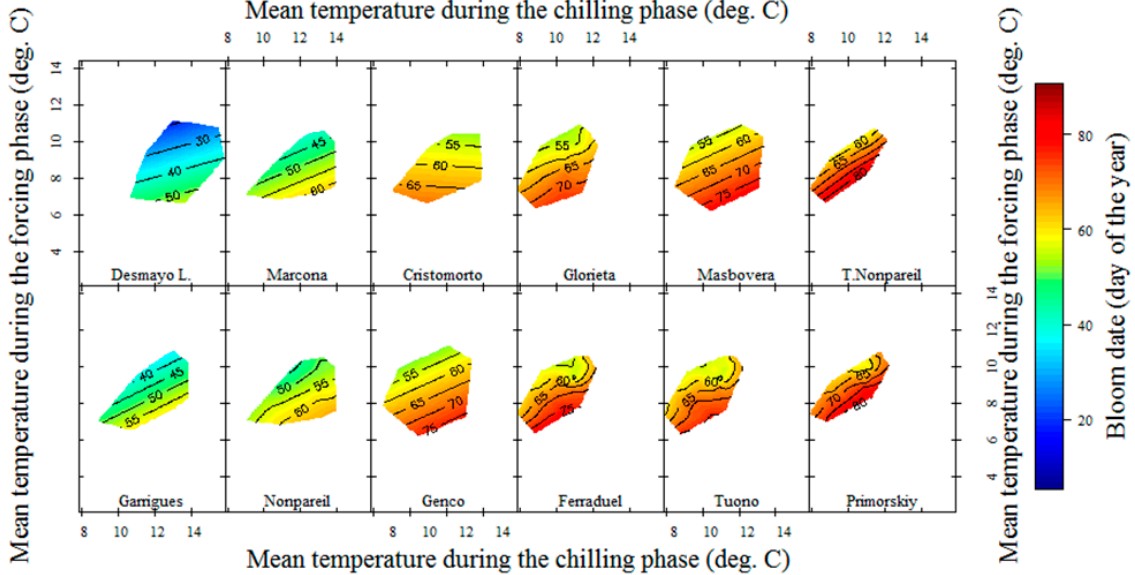

**Figure 7.** Response of 'Desmayo Largueta', 'Marcona', 'Cristomorto', 'Glorieta', 'Masbovera', 'Tardy Nonpareil', 'Garrigues', 'Nonpareil', 'Genco', 'Ferraduel', 'Tuono', and 'Primorskiy' blooming dates to mean temperatures during the chilling and forcing phases in Mas de Bover for the period 1979–2015. Isolines connect homogeneous blooming dates in this space. The color spectrum has to be interpreted as variation of the flowering dates. Individual plots can be found in Supplementary Figures S35–S59.

If isolines are relatively parallel to the X-axis, this implies that variation of mean temperatures during the chilling period did not influence the advancements or delays of flowering dates (blooming date did not change irrespective of the mean temperature during the chilling phase).

Instead, these nearly horizontal isolines show that changes in the mean temperature during the forcing phase would influence blooming, as a change in the vertical axis implies a change of isoline. This can be seen in Figure 7 for 'Desmayo Largueta', 'Cristomorto', and 'Marcona'. On the other hand, some cultivars like 'Tardy Nonpareil', 'Masbovera', and 'Primorskiy' (Figure 6) showed isolines with clear slope, meaning that temperatures on both chilling and forcing phases influenced the interannual blooming date variability.

Early-blooming cultivars 'Desmayo Largueta' and 'Garrigues' and mid-blooming cultivars 'Marcona' and 'Nonpareil' showed a stronger bloom-advancing effect of warm mean temperatures during the forcing phase than late and extra-late flowering cultivars 'Primorskyi' and 'Tardy Nonpareil' (Figure 7). Moreover, the latter presented contour lines with a clear slope, meaning that the behavior of their past blooming dates could be influenced almost equally by mean temperatures during both chilling and forcing phases

Another important aspect to analyze is the sensitivity of each cultivar to small variations on mean temperatures during both phases. This sensitivity could be perceived by observing the distance between isolines, the higher the distance between isolines the lower the sensitivity to small variations of temperature. In this case, some cultivars showed a higher sensitivity, e.g., 'Garrigues', 'Ferraduel', 'Tuono', 'Primorskiy', 'Garbí', and 'Tardy Nonpareil' (Figure 7), while others remained more impassive (or less sensitive) to those small variations, e.g., 'Desmayo Largueta' and 'Cristomorto' (Figure 7). All these patterns can also be seen in other cultivars in Supplementary Figures S35–S60.

### 3.4.2. Apple Cultivars

For most apple cultivars, changes in mean temperatures during both chilling and forcing phases influenced the year-to-year variation on their blooming dates but 'Brookfield Gala', 'Golden Smoothee', and 'Jeromine' showed the most marked effect (Figure 8). However, the nearly horizontal isolines presented for 'Aporo', 'Red Chief', and 'Pink Lady' (Figure 8) show a higher dependence on warm mean temperatures during the forcing phase for the advancement of the blooming dates.

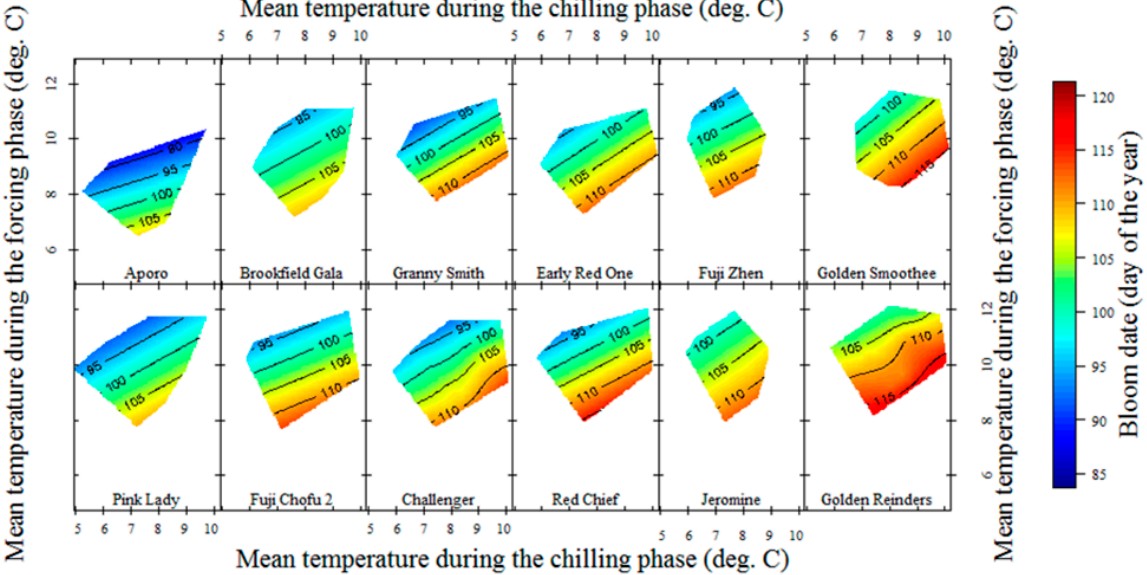

**Figure 8.** Response of 'Aporo', 'Brookfield Gala', 'Granny Smith', 'Early Red One', 'Fuji Zhen', 'Golden Smoothee', 'Pink Lady', 'Fuji Chofu 2', 'Challenger', 'Red Chief', 'Jeromine', and 'Golden Reinders' blooming dates to mean temperatures during the chilling and forcing phases in Mas Badia for the period 1992–2018. The color spectrum has to be interpreted as variation of the flowering dates. Individual plots can be found in Supplementary Figures S61–S72.

As for grade of sensitivity to temperature variations, the majority of the apple cultivars presented relatively distant isolines (Figure 8), meaning a low sensitivity to small variations of the temperature in both chilling and forcing periods. The highest sensitivity to changes in mean temperatures was observed in 'Red Chief' and the lowest in 'Brookfield Gala' and 'Jeromine' (Figure 8).

## 4. Discussion

### 4.1. Chilling and Forcing Phases Delineation Using a Partial Least Squares (PLS) Approach and Its Limitations

The identification of the chilling and forcing phases was quite simple using the PLS analysis as implemented in the chillR package [2,36]. The start date of the chilling phase was clear and similar for all almond and apple cultivars, and we should stress that the same criterion was used for both species, i.e., a stable sequence of negative coefficients, as described in the materials and methods section.

A large difference in the chilling phase end dates was observed between the different groups of almond cultivars, which suggests that the later the blooming date, the later the completion of the chilling phase. Similar results were found by Benmoussa et al. [23]. Furthermore, El Yaacoubi et al. [15] found that low-chill apple cultivars showed very early flowering dates compared to high-chill cultivars. However, as mean bloom dates did not differ much between the 12 apple cultivars analyzed herein, no clear relationship could be found between the CR or the length of the chill/forcing phase and their bloom dates like those we found for almond.

Despite the presence of overlaps and gaps, the beginning of the forcing phase seems to be related to the end of the chilling phase. This finding could be interpreted as an interdependency of the two processes. In fact, notwithstanding the different methods used, the start date of the forcing phase is similar in this work and in Funes et al. [12], which assumed that the beginning of the forcing phase matched the end of the chilling phase. As those cultivars come from breeding programs that looked for a specific blooming time (e.g., late bloomers in almond), the flowering date is somewhat externally fixed. Hence, the longer the phases, the higher the probability that they overlap. Likewise, the shorter the phases, the lower the probability of an overlap and the more likely a gap between phases. As for the overlapping period observed in several late and very-late almond cultivars and in the apple cultivar 'Aporo', Harrington et al. [6] and Campoy et al. [44] suggested that this phenomenon could be produced after a tree's critical CR has been met. In the present study, this overlap is found when 50% or more of their CR has been fulfilled and after 86% in the case of the apple cultivar 'Aporo'.

The presence of this overlapping period may be attributed to parallel or overlapping models being more adequate to describe blooming temperature dependences [6,11]. However, while giving a biological basis for overlapping [45,46], none of these models present a clear base for gaps as detected here in several almond and almost all apple cultivars. Both overlaps and gaps could also be the result of the inadequacy of the method used in this study to clearly delineate both phases. As a matter of fact, despite the fact that the start date of the chilling phase was clear and similar for all almond and apple cultivars, the presence of a pattern characterized by periods with low VIP values and positive model coefficients made the delineation of the chilling phase end less clear, especially in the almond late bloomers and for 'Jeromine' and 'Golden Smoothee' apple cultivars. The same pattern has previously been observed for almond and pistachio [23,24], walnut [2,28], cherry [2,41], chestnut and jujube [26], and apricot [25]. Guo et al. [26] concluded that, probably, the continuous accumulation of chill throughout the chilling phase is a simplification of a more complex chilling accumulation process. Although this may be the case, the presence of a significant number of chill portions accumulated in these periods, both in our study and in Guo et al. [26] and references therein, weakens this explanation and, if true, it would imply, at least, discounting those periods from chill accumulation. Yet, Benmoussa et al. [23] proposed a physiological base for this behavior: they suggested that discontinuities in chill/heat accumulation could be related to different physiological processes inside the bud that rely on a genetic basis, like the variation of DAM6 gene expression during peach dormancy [47]. Another alternative explanation related to the method used to delineate the chilling

phase would be that PLS, when used to delineate the chilling phase, is not really able to establish the relevance of every single day in the chill accumulation, even it is aimed to it.

Reasons for this limitation might be: (i) the distance to the blooming day (higher than for the forcing phase), (ii) the presence of periods in which chill and heat accumulation co-occur (overlapping periods), or (iii) the high autocorrelation between temperatures in consecutive days, reinforced by the running mean used, making the information enclosed in these days irrelevant to define the PLS model (hence, low VIP values), even if PLS is designed to deal with this situation. A result that supports this limitation of the method is the presence of days with high VIP values and negative coefficients during the heat phase, after the date of blooming. As the influence on a past event is not to be considered, this probably reflects a high correlation with days before the blooming date. Moreover, if the end date of the chilling phase is not accurate, the observed overlapping of chilling and heat phases may be at least partially artifactual. In summary, this alternative explanation would mean that the method used in this study is not sensitive enough to determine the end of the chill phase.

All in all, the determination of the end of the chilling phase as well as the delimitation of the forcing phase are crucial as the length of the chilling and forcing phases determine the total amount of CP and GDH accumulated. Thus, delays in the choice of the end of the chilling phase imply an overestimation of chill accumulation [48]. Moreover, the relationship between both phases needs to be determined if estimations of future blooming dates are to be obtained from climatic projections. For this reason, further efforts must be carried out to investigate the transition of these two phases as well as how these phases occur during plant dormancy to improve the characterization of horticultural fruit trees using their CR and HR.

Despite the limitations pointed out in this section, we do believe that PLS, as used in the chillR package, is a good tool for chill and heat phases delineation, and the reasons are given in the next section.

## 4.2. Chill and Heat Accumulation

### 4.2.1. Chill and Heat Requirements for Almond

CR and HR for almond have previously been estimated in several studies [15,23,48–51]. However, the lack of standard methods to calculate these agroclimatic requirements hinders the possibility of an accurate comparison with our results. Benmoussa et al. [23], using the PLS method with the dynamic model in Sfax (Tunisia), obtained similar CR results despite the warmer climate in Sfax for most of the eight cultivars also appearing in our study: 'Tarragona', 'Cristomorto', 'Nonpareil', 'Genco', 'Ferragnes', 'Cavaliera', 'Ferraduel', and 'Tuono' (although the last two show very different CR). This consistency of CR estimations based on the use of PLS analysis despite the different climate and different authors and chillR users implied in the estimations, reinforces the validity of the method, despite the criticisms exposed in the previous section.

Ramírez et al. [49], who shares a similar Mediterranean climate (Central Valley of Chile) and the use of the dynamic model with our study, calculated a CR for 'Desmayo Largueta', 'Ferragnes', and 'Nonpareil' with variable results: the former two presented moderate differences with our results, but the CR calculated for 'Nonpareil' is comparable in both studies. On the other hand, El Yaacoubi et al. [15] calculated the CR accomplished in two consecutive phenological seasons in the Northern Morocco using the 'single-node cutting' method and the Tabuenca test, showing strong divergence with the estimations herein for 'Ferragnes', 'Tuono', and 'Marcona'. An explanation for the differences between these results is the different climate where almond trees were cultivated, although Benmoussa et al.'s [23] results for 'Ferragnes' were similar to ours, and thus different from El Yaacoubi et al. [15], with whom climate is shared. We believe that the use of different methods (natural condition-statistical versus forced conditions-experimental) to determine the duration of the chilling phase may be the cause of the difference. The case of 'Tuono' seems to be singular as the results obtained by Benmoussa et al. [23] and El Yaacoubi et al. [15] are similar and quite different from ours, but Gaeta et al.'s [50] results in Apulia (Italy) are similar to our estimations, which might be due to the presence of two different

local sub-varieties of the cultivar, as Apulia and Catalonia share similar climate conditions. They also found results for 'Cristomorto' and 'Rana' that resulted similar to ours. Although we cannot explain every discrepancy with the literature, our results mostly agree with comparable studies, and altogether, we believe that they are consistent and reliable.

The same applies to HR: the results obtained for the cultivars 'Ramillete', 'Marcona', 'Rof', 'Nonpareil', 'Texas', 'Cristomorto', 'Masbovera', 'Ferragnes', 'Ferraduel', 'Tuono', 'Primorskiy', 'Rana', 'Tarragonès', and 'Genco' reasonably matched those obtained in the studies of Alonso et al. [39], Egea et al. [48], Benmoussa et al. [23], and Gaeta et al. [50]. However, the cultivars that had elevated CR together with low HR (e.g., 'Texas') and vice versa (e.g., 'Cristomorto'), did not follow the same pattern in those studies. Similarly, 'Primorskiy', which presented high CR and HR, followed a different pattern in Alonso et al. [39], where it showed low CR together with high HR. This opposing behavior of 'Primorskiy' might seem contradictory, as the flowering date is very similar in both studies, but the overlapping we estimated (42% of the chill plus heat accumulation period) might explain it. In fact, the sequential method used in Alonso et al. [39], which first estimates a forcing phase and then attributes the remaining days to the chilling phase, should result in large underestimations of CR in cultivars like Primorsky, provided the overlap estimated here is not artifactual.

Almond shows the widest range of blooming dates between all the fruit and nut species [52]. In this study, in contrast with Alonso et al. [39], a considerable variability in almond CR was observed, while the variability in heat requirements were smaller (Table S4). These results suggest that the main trait defining early to late blooming cultivars in a Mediterranean climate is their CR. In accordance with our results, Gaeta et al. [50] obtained higher coefficients of variation on CR than on HR, suggesting that the flowering period is more dependent on CR in warm climates. Egea et al. [48] also reported CR as the main factor differentiating early from late blooming cultivars. They conducted the study in South-East Spain, with a mild climate more similar to Mas de Bover than to the colder Ebro River Central Valley, where Alonso et al. [39] worked.

However, while this is the general situation, it is not always the case, e.g., 'Ferragnes' and 'Ferraduel' present a similar bloom date; however, they differed significantly both in their CR and HR. It seems that both requirements compensated for each other to result in similar bloom dates. The interesting thing is that these are sister cultivars, which indicates that it should not be too difficult to breed new late blooming cultivars based in high HR, not high CR. In fact, Alonso Segura et al. [53] discuss that an indirect selection for high heat requirements might have taken place while breeding for late blooming for central Asia.

### 4.2.2. Chill and Heat Requirements for Apple

The estimated CR and HR for 12 apple cultivars in Mas Badia were compared to Funes et al. [12], where the same cultivars were used (although up to five years of data and some cultivars were added in this study). Despite the fact that the HR results agree with the former study, we found substantially lower CR estimates. For the same cultivars, we obtained a range of 37.79 to 50.26 CP, whereas Funes et al. [12] ranged from 62.5 to 68.4 CP. Since the raw data are basically the same, this divergence may be linked to the use of different methods to delineate the chill/forcing phases. In this case, Funes et al. [12] used a sequential chill-forcing model [39], where chill and heat accumulate consecutively, as explained above. Thus, in comparison with our study, three different dates needed to be fixed instead of four. This gave place in our case to a gap in most cases, that is absent in Alonso et al.'s [39] approach. Moreover, chill accumulation was calculated from 1 October in Funes et al. [12], while in this study, the PLS approach resulted in chill accumulation starting about mid-November. Nevertheless, our CR and HR estimations for the apple 'Granny Smith' agree with El Yaakoubi et al. [15], in contrast with almond results.

Even though the coefficient of variation was greater for CR than for HR, the difference between these estimates was narrower than for almond, probably because, in contrast to almond, this set of apple cultivars do not share a specific breeding interest to obtain late or early blooming cultivars.

In this cultivar collection, the effect of chill and heat defining the flowering date seem to be similar, thus, different combinations of chill and heat would define the same bloom dates in different cultivars.

*4.3. Past Blooming Trends and Impacts of Chilling and Forcing Temperatures on Blooming Dates*

Despite an increasing trend in maximum and mean annual temperature in Mas de Bover, no significant trends were observed in blooming during the last 37 years for the 25 almond cultivars. The reason for this lack of effect might be a compensation between the higher time needed to obtain CR and the lower time to reach HR [12]. Similar results were observed for the 12 apple cultivars in Mas Badia, where no blooming trends were observed, maybe related to the lack of statistically significant past temperature trends for the last 26 years [12].

Three-dimensional plots (Figures 6 and 7) show the relationship between the CR and HR of a given cultivar and the climate of the location where it is grown: cultivars with low CR, grown in locations even with mild winters, as in this study, are expected to present low influence of mean temperature along the chilling phase as they will easily accumulate enough chill even in the case of relatively high winter temperatures, hence presenting quite horizontal isolines in the three-dimensional plots. In contrast, for the same locations, cultivars with high CR will show the influence of the temperature both during the chilling and the heat phase, presenting isolines with higher slopes. Finally, a cultivar with high CR and low HR, or growing in a warmer location, will be very influenced by changes in temperature during the chilling phase, but will easily achieve its HR, so showing low influence of the temperature during the heat phase, and thus showing quite vertical isolines. When analyzing the impacts of chilling and forcing temperatures on the bloom dates for almond, the nearly horizontal contour lines produced for most cultivars (Figure 6 and Supplementary Figure S60), indicate a higher effect of mean temperatures during the forcing phase on advancing and delaying past blooming dates. By contrast, for most apple cultivars, changes in mean temperatures during both chilling and forcing phases influenced the year-to-year variation in their blooming dates (Figure 7).

## 5. Conclusions

Different CR and HR were obtained for 25 almond and 12 apple cultivars, making this study an exhaustive analysis of the thermal requirements for these species. The considerable variability on CR between almond cultivars was interpreted as the main trait defining early to late blooming cultivars. For apple, different combinations of chill and heat would define the same bloom dates in different cultivars.

The relative importance of mean temperatures during chilling and forcing phases on the intravarietal flowering dates depends on the cultivar and the climatic conditions of the place where fruit trees have been cultivated. Using past blooming dates trends, we found that under a Mediterranean climate, advancements and delays on almond phenology will be produced by changes in mean forcing temperatures. By contrast, for apple, mean temperatures in both chilling and forcing phases will influence the variation of the blooming dates for this species in the future.

Despite its limitations, the PLS analysis has proven to be a useful tool to define both chilling and forcing phases for the subsequent calculation of almond- and apple-specific CR and HR. The consistency of CR and HR estimations based on PLS use with previously published results despite the different climate and different people implied in the estimations, reinforces the validity of the method. Nevertheless, since the delineation of these phases determines the total amount of CP and GDH accumulated, further efforts must be carried out to investigate the transition of these phases as well as how these phases occur during plant dormancy to improve the characterization of horticultural fruit trees.

The findings in this study might be useful for future research regarding climate change impacts on deciduous fruit phenology and feasibility in the Mediterranean context. Consequently, our results might also be valuable for climate adaptation strategies ensuring CR fulfillment of temperate fruit trees in a global warming context.

**Supplementary Materials:** The following are available online at http://www.mdpi.com/2073-4395/9/11/760/s1; Figure S1: Annual temperature trends (mean, maximum and minimum) in Reus Airport during the period 1978-2015; Figures S2 to S24: Results obtained from the PLS regression analysis between blooming dates and daily mean chill and heat accumulation in Mas de Bover using the dynamic model and the GDH model for the 23 almond cultivars not shown in Figure 3; Figures S25 to S34: Results obtained from the PLS regression analysis between blooming dates and daily mean chill and heat accumulation in Mas Badia using the dynamic model and the GDH model for 10 apple cultivars not shown in Figure 4; Figures S35 to S59: Individual plots of the response of blooming dates to mean temperatures during the chilling and forcing phases in Mas de Bover for all almond cultivars studied; Figure S60. Response of blooming dates to mean temperatures during the chilling and forcing phases in Mas de Bover for 'Cavaliera', 'Gabaix', 'Ardechoise', 'Alicante', 'A-258', 'Ferragnes', 'Garbí', 'Ramillete', 'M.Tarragona', 'Rof', 'Rana', 'Texas', and 'Tarragones'; Figure S61 to S72: Individual plots of the response of blooming dates to mean temperatures during the chilling and forcing phases in Mas Badia for the apple cultivars studied; Table S1: Temperature details during the phenological seasons in Mas Badia and Mas de Bover; Table S2: Blooming past trends for 25 almond cultivars studied in Mas de Bover; Table S3: Blooming past trends for 12 apple cultivars studied in Mas Badia; Table S4: Temperature past trends for the studied period in Reus Airport WS (1978–2015) and La Tallada WS; Table S5: Analysis of Variance and Duncan's multiple range test for bloom date of almond blooming groups; Table S6: Analysis of Variance and Duncan's multiple range test for blooming dates (DOY; day of the year) of the 25 almond cultivars; Table S7: Analysis of Variance and Duncan's multiple range test for blooming date (DOY; day of the year) of the 12 apple cultivars; Table S8: Analysis of Variance and Duncan's multiple range test for bloom date of almond blooming groups plus apple cultivars; Table S9: Analysis of Variance and Duncan's multiple range test for chill requirements (CR) in chill portion (CP) of almond blooming groups; Table S10: Analysis of Variance and Duncan's multiple range test for heat requirements (HR) in growing degree hours (GDH) of almond blooming groups.

**Author Contributions:** Conception and work design: R.S., X.A., C.B. and F.d.H.; Supervision: X.A. and I.F.; Meteorological data acquisition and compilation: I.F. and C.B.; Methodology (Reconstructing temperature series): X.A., I.D.-P. and C.B.; Phenology data acquisition, analysis, and compilation: X.M., F.V., G.À., J.C. and C.B.; Methodology (statistical analysis): I.D.-P., X.A. and I.F.; Figures and Tables edition: I.D-P. and I.F.; Project Administration: C.B. and F.d.H.; Funding acquisition: R.S. and C.B.; Original draft preparation: I.D.-P.; Writing-Review and editing: I.D.-P., X.A. and I.F.; Article review: all authors.

**Funding:** This research was partially funded by the LIFE programme of the European Commission through the project "LIFE12ENV/ES/000536-Demonstration and validation of innovative methodology for regional climate change adaptation in the Mediterranean area (LIFE MEDACC)".

**Acknowledgments:** We want to thank the work of the IRTA's technical personnel in Mas de Bover and Mas Badia in phenological stages recording during the last 40 years that has been critical for this study. We also want to thank the Spanish (AEMET) and the Catalan (SMC) Meteorology Agencies for providing the long-term series of temperature data needed to carry out this study. We deeply thank Segio Vicente-Serrano and Javier Zabalza from Instituto Pirenaico de Ecología—Consejo Superior de Investigaciones Científicas for their help in meteorological data acquisition from AEMET and in reconstructing temperature series.

**Conflicts of Interest:** The authors declare no conflict of interest.

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
