# Peer review of "Blooming under Mediterranean Climate: Estimating Cultivar-Specific Chill and Heat Requirements of Almond and Apple Trees Using a Statistical Approach"

_agronomy, doi:10.3390/agronomy9110760_

Round 1
Reviewer 1 Report
Diez-Palet et al. Have addressed most of my comments. In general, these corrections have improved the manuscript. However, few improvements can still be done. Particularly, I would encourage the authors to make it consistent between what they have responded to the reviewers and what they have actually addressed in the main manuscript. Some minor comments below.
L18 and L74. Make it consistent the use of CP or CPs for Chill Portions
L31. Consider to change “ChillR” to “chillR”
L38. Add a space between “from” and “the”
L55. Remove the additional period
L64. Make it consistent between 677,328ha and 43,123 ha (L63)
L64. In my first report, I have mentioned about specifying the relevance of almonds by using also percentages. In author’s responses to reviewer, they responded to that comment as “Done”. However, they did not addressed the comment as they stated. Please clarify.
L84. One of my comments was about citing chillR in this line. The authors responded: “It is in the appropriate section”. Ok, but why they decided to cite the PLS procedure and the R environment and not the package? What is the “appropriate section” then? To me, the right place is the first appearance
L150. My point on this issue is that the reference [37] is for chillR version 0.70.2 but in the text, the authors say they used version 0.62. This is not consistent
L222. Authors did not modify the headings as they say in the document of responses to reviewer
Table 1 and continuation. First row for the varieties is still in other format
Author Response
L18 and L74. Make it consistent the use of CP or CPs for Chill Portions
L31. Consider to change “ChillR” to “chillR”
L38. Add a space between “from” and “the”
L55. Remove the additional period
L64. Make it consistent between 677,328ha and 43,123 ha (L63)
All previous remarks, checked and changed
L64. In my first report, I have mentioned about specifying the relevance of almonds by using also percentages. In author’s responses to reviewer, they responded to that comment as “Done”. However, they did not addressed the comment as they stated. Please clarify.
From this comment, we guess this corresponds to Reviewer 2 in the first version of the manuscript. We are sorry for the confusion: Reviewer 1 made a similar comment and we gave there our reasons to keep figures as they were. We reproduce it here: “Here we are trying to show the relevance of both crops in Catalonia and Spain. In the case of almonds, what is relevant is the amount of almonds produced, not the percentage over Spanish production, which is world’s third largest. In the case of apples, Spanish production is not son relevant in the world, but Catalan apple production does in Spain’s context. That is the reason to present these figures in the way we do.”
We would like to add that, with the absolute numbers provided, getting a % is easy for the reader.
L84. One of my comments was about citing chillR in this line. The authors responded: “It is in the appropriate section”. Ok, but why they decided to cite the PLS procedure and the R environment and not the package? What is the “appropriate section” then? To me, the right place is the first appearance
There was a misunderstanding from our side here: now we understand the referee meant to introduce reference 37 here and not in materials and methods. We thought he/she meant to give package number here, which we believe belongs to materials and methods. We have changed the reference
L150. My point on this issue is that the reference [37] is for chillR version 0.70.2 but in the text, the authors say they used version 0.62. This is not consistent
This is true: what we tried to explain is that the compiled Manual for version 0.62 is no longer available as R packages continuously evolve and are substituted in the repository. Although past versions can still be found, past compiled past manuals don’t, so there was no way to give a reference for the packaged used at the moment, only pointing to the repository of older versions. We considered it was better to give the link to the presently available package. However, we agree with the reviewer this might be misleading, so we have reworked the reference adding a links both to the present version and to the 0.62 version in the old versions repository.
L222. Authors did not modify the headings as they say in the document of responses to reviewer
The reviewer is completely right: we did not change the subheadings. We should have written “checked” and not “changed” throughout. The reason not to change it is that we followed the template for the special issue, and gave to the headings of the subsubsections the format that appears in the template, as follows:
Results
This section may be divided by subheadings. It should provide a concise and precise description of the experimental results, their interpretation as well as the experimental conclusions that can be drawn.
3.1. Subsection
3.1.1. Subsubsection
Table 1 and continuation. First row for the varieties is still in other format
We missed that. Changed.Thank you

Reviewer 2 Report
It appeared from the authors' response that, based on the request from reviewer 1, a more sound physiological basis was going to be given:
"In fact, reviewer 1 asked for more sound physiological basis for gaps."
The authors still kept the same sentence without change that mentions a single gene that may be involved in the process. There is nothing in their introduction regarding gene regulation of dormancy or budbreak.
Other comments that may have required additional work by authors were disregarded or brushed off with little justification besides disagreement. Therefore, I don't believe there is anything I can add by further reviewing this manuscript and no additional comments will be provided.
Author Response
We are sorry this reviewer (with seems to be reviewer 3 in our previous submission) does not accept our answers to his/her comments. She/he complains that we give little justification to disregard some of the changes. But we have no clue from her/his side to which points she/he refers to or what is missing in our answers.
In relation to the gene expression subject, we have reviewed again how it is expressed in the manuscript, and we believe it is clearly expressed as a possible explanation not derived from our results. Moreover, the next sentence proposes that our result can be a caused by the methodology used, and later on this is even defined as possible artifactual.
We mentioned reviewer 1 in contrast with reviewer 3 in the sense that one seemed to ask more and the other one less. Certainly, we did not add more physiological basis as we are not aware of any.
All in all, we believe we have considered and answered his/her comments on this point

This manuscript is a resubmission of an earlier submission. The following is a list of the peer review reports and author responses from that submission.
Round 1
Reviewer 1 Report
Overall, this paper was thoroughly researched, and data analysis was well-executed and well-explained. My major reservation with recommending this paper is that it is not clear how this work is novel or moves the questions of how we quantify chill and heat forward. Authors used a dataset that had, to a certain extent, already been used, and a statistical approach for quantifying chill and heat requirements that had already been used. They do a good job of comparing their results with similar works (same crop, different approach and different crop, same stats approach) but they never come to a conclusion about what these comparisons mean. That last step is critical to differentiating a report that characterizes a germplasm from a scientific journal article. They say their estimations of chill and heat requirements are different, but are they more or less valid/accurate/revealing than attempts by other researchers? They say their results show some weakness in the PLS approach, but don’t take a stance on whether that makes them skeptical of their results, or this approach as a whole, or whether they believe (based on x, y, z reason), that despite the weaknesses revealed, PLS is still the best approach. Put another way, there were so many papers on almond and apple chill and heat requirements before this one, and numerous papers on using Guo-Leudeling’s PLS approach to quantify chill and heat requirements. What question did the authors feel had not been answered by those papers that needed to still be answered by their investigation? Why is this paper necessary?
More specific comments:
Citation 11 modeled temperatures, not phenology. This would be more appropriately grouped with next citiation. “One of the most…” This is a bold, unsupported and unnecessary statement.
61-64. Long, awkward sentence. Suggest splitting into two sentences at “a significant part”.
28,256 ha
64-66. Consider consistently using the percent of total Spanish supply for both almonds and apples.
Don’t need both in-shell and without shell numbers, but be clear which you are giving.
65-66. Given the Catalonian almond supply is ~5% of Spanish total, seems irrelevant that Spain is 3rd in global almond supply.
74-76. Bring up context of climate change as a driver for this research, but this research doesn’t put the CR found in context of future anticipated CRs. Consider Luedeling’s global CR projections for this.
106-108. Explain if/how Funes’s analysis was different from present analysis.
Are author’s concerns about using airport temperatures? Particularly the impact of radiative heat at night from paved surfaces influencing Tmin. Could seriously skew the data relative to temperatures experienced in the field.
124, 128-129. Better to give p values in the results than list different levels of significance here.
135-138. That was not the point of Darbyshire or Campoy. Better to cite 19 & 20 again.
138-140. This sentence can only be supported by papers that compared multiple ways of counting chill. 26 did not.
This was not the point of Guo’s paper. Just cite 21.
150-151. Why cite Legave? Why not cite original PLS chill-heat papers?
“chilling and forcing” “established”
168-169. Unclear what statistical reasons are, what is meant by “independently”.
“until”
172-174. Does it concern researchers that forcing phase extended beyond mean bloom data in almost every case? Does this not seem to reveal a flaw/weakness in the PLS approach? If not, why not? Same re: 232-233.
208-213. Why follow the classifications of Vargas and Romero when you have more than two times as much data as them, and your results contradict them? Why not make your own groupings of early-mid-late?
Oct 23-24 is, roughly speaking, the average start found by the PLS analysis. That doesn’t mean it is truly when the trees/buds started counting each year.
235-238. What do the authors think is the significance of this lack of clear delination?
251-255. It is not clear how overlaps in counting or gaps in counting is shown in the figures.
161-262. But more overlap in mid-blooming when CR and HR was same as early blooming (e.g. Ramillete vs. Mollar, Rof, Alicante, Rana). This seems more interesting and somewhat new to me.
What does “longed up” mean?
Table 1. This table is very hard to read with the wrapping text. Consider switching to landscape so all information for one cultivar is on one line.
“months” “established”
Figure 4. Why aren’t VIP bars blue and green?
No 3.4 Heading? Just skip to sub-heading?
Figure 5. Would be interesting to overlay bloom timing on this figure. Maybe color almond dots by earl/mid/late bloom.
“X-axis, this implies” Use “Instead” rather than “On the contrary” “horizontal” To my eye, Cristomorto is the only cv. without a clear slope. If you’re going to differentiate, you need to give slope numbers, not just your visual assessment.
371-376. Give summary statistics to support this assertion/differentiation.
377-383. Give summary statistics. Don’t just rely on visual assessment.
390-394. Need to put numbers to the slopes. Doesn’t look “nearly horizontal” to me.
402-404. Need regression statistics to support this. Chill phase end date vs. bloom date.
23 or 22? I think you mean 22.
404-406. Do you really mean “On the other hand”? Isn’t this essentially the same = Late chill met à Late flowering, Early chill met à Early flowering
409-420. How does this fit into the context of chill overlap work in almonds and apples of Pope et al (2014), Pope et al (2017) and Darbyshire et al (2017)?
409-451. I see potential glimmers of how this work is novel in these paragraphs, but authors would need to make it more clear, and emphasis in the abstract, title and introduction.
437-439. Is another possible explanation that the overlap is because the CR and HR are not consistently met on the same day in each year of the record?
458-531. Would be valuable, efficient and easier for the reader if you have a table of your estimates CRs and HRs, compared with the CR and HR estimates for the same cultivars from other works, e.g Benmoussa, Ramirez, El Yaccoubi. Rather than making the reader take your word for it that the numbers are similar or different, or taking up space listing the numbers, or making the reader go hunt down the numbers themselves to confirm what you are saying.
478-479. OK, different approaches yielded different results. But what numbers do you believe more and why? Given that El Yaacoubi was using a controlled experiment and PLS is just empirical, does this cast doubt on empirical findings? Is there a rational way to meld the two results together?
Awkward wording. Define “this”.
519-522. What is the physiological basis for including this “gap” in the model as a possibility? Do you think your results are more true to the real world experience of the trees/the true mechanism of chill-heat than Funes? And why?
Alonso or Funes? I think you mean Funes here.
554-592. This section doesn’t integrate or analyze or contextualize any of the data from this paper. Either bringin in data and show what it means in the light of these predicted temperature increases, or delete whole section.
715-717. Reformat all caps.
Author Response
Overall, this paper was thoroughly researched, and data analysis was well-executed and well-explained. My major reservation with recommending this paper is that it is not clear how this work is novel or moves the questions of how we quantify chill and heat forward. Authors used a dataset that had, to a certain extent, already been used, and a statistical approach for quantifying chill and heat requirements that had already been used. They do a good job of comparing their results with similar works (same crop, different approach and different crop, same stats approach) but they never come to a conclusion about what these comparisons mean. That last step is critical to differentiating a report that characterizes a germplasm from a scientific journal article. They say their estimations of chill and heat requirements are different, but are they more or less valid/accurate/revealing than attempts by other researchers? They say their results show some weakness in the PLS approach, but don’t take a stance on whether that makes them skeptical of their results, or this approach as a whole, or whether they believe (based on x, y, z reason), that despite the weaknesses revealed, PLS is still the best approach. Put another way, there were so many papers on almond and apple chill and heat requirements before this one, and numerous papers on using Guo-Leudeling’s PLS approach to quantify chill and heat requirements. What question did the authors feel had not been answered by those papers that needed to still be answered by their investigation? Why is this paper necessary?
About the novelty of the paper, we call to the attention of the reviewer that there are not previous studies for this number of cultivars in this specific regions, and to our knowledge this is the first time they are presented together. These species in this specific area had not been. This is recognized by Reviewer 2 and, to some extent, Reviewer 3. In addition, this reviewer recognizes some novelty in part of the discussion related to PLS use and results. We agree that this part was hidden in the original manuscript, so we have reflected it in the Abstract and the conclusions.
Only a part of one of the datasets had been used before, but it has been enlarged both in length and in number of cultivars, and a different statistical approach has been used.
We believe we have improved the Discussion in relation to the comparison with the literature, although we have not followed all the indications of the reviewer. Our reasons appear below in response to his/her remarks. Specifically, we believe that the comparison with the literature reveals that our results are consistent. In specific discrepancies, we cannot give a final conclusion on whether our results are better or worse than others, but as a whole we believe they are consistent and reliable.
In relation to PLS validity, the manuscript already states in section 4.2.1 that PLS results are valid despite our criticisms, and also in the conclusions. We have, however, reinforced the statement in the conclusions and added this conclusion to the abstract to make clear that, despite possible limitations described here, PLS approach is valid and a good tool for CR and HR analysis.
More specific comments:
Citation 11 modeled temperatures, not phenology. This would be more appropriately grouped with next citiation.
Done
“One of the most…” This is a bold, unsupported and unnecessary statement.
We strongly disagree: this is not at all unsupported, and the statement is necessary to justify a context in which the interest of our results increases.
It is true that the reference we provided is wrong as in the Synthesis report, WGI refers in general to dry regions, or dry subtropical regions among which the report includes the Mediterranean. We should have cited the complete report of the WGI. However, this report is more related to the magnitude of the effects of climate change that to its vulnerability. Hence, and following also comments of Reviewer 3, we have changed the reference to the IPCC Special Report Global warming of 1.5°C and specifically to Chapter 3 Impacts of 1.5°C of Global Warming on Natural and Human Systems.
We have also changed our statement to specify than the Mediterranena is one of the most affected and vulnarable regions to Climate Change.
Below, some references to support this claim, both from the IPCC WGI report and form the IPCC Special Report Global warming of 1.5°C
IPCC Special Report on the impacts of global warming of 1.5°C above pre-industrial levels.
Chapter 3: Impacts of 1.5°C of Global Warming on Natural and Human Systems
3.5.4.5 Southern Europe and the Mediterranean
The Mediterranean is regarded as a climate change hotspot, both in terms of projected stronger warming of the regional land-based hot extremes compared to the mean global temperature increase (e.g., Seneviratne et al., 2016) and in terms of of robust increases in the probability of occurrence of extreme droughts at 2°C vs 1.5°C global warming (Section 3.3.4). Low river flows are projected to decrease in the Mediterranean under 1.5°C of global warming (Marx et al., 2018),with associated significant decreases in high flows and floods (Thober et al., 2018), largely in response to reduced precipitation. The median reduction in annual runoff is projected to almost double from about 9% (likely range 4.5–15.5%) at 1.5°C to 17% (likely range 8–25%) at 2°C (Schleussner et al., 2016b). Similar results were found by Doll et al. (2018). Overall, there is high confidence that strong increases in dryness and decreases in water availability in the Mediterranean and southern Europe would occur from 1.5°C to 2°C of global warming. Sea level rise is expected to be lower for 1.5°C versus 2°C, lowering risks for coastal metropolitan agglomerations. The risks (assuming current adaptation) related to water deficit in the Mediterranean are high for global warming of 2°C but could be substantially reduced if global warming were limited to 1.5°C (Section 3.3.4; Guiot and Cramer, 2016;Schleussner et al., 2016b; Donnelly et al., 2017). P259
Table 3.6 p 261
(…) the Mediterranean is an example of a region with high vulnerability where various adaptation responses have emerged. Previous IPCC assessments and recent publications project regional changes in climate under increased temperatures, including consistent climate model projections of increased precipitation deficit amplified by strong regional warming Box 3.2 p 200
(…) important risk of extreme drought conditions for the Middle East under 1.5°C of global warming with risks being even higher in continental locations than on islands; these projections are consistent with current observed changes. Risks of drying in the Mediterranean region could be substantially reduced if global warming is limited to 1.5°C compared to 2°C or higher levels of warming
Higher warming levels may induce high levels of vulnerability exacerbated by large changes
in demography. Box 3.2 p 200
Warmer and drier conditions in particular facilitate fire, drought and insect disturbances, while warmer and wetter conditions increase disturbances from wind and pathogens (Seidl et al., 2017). Particularly vulnerable regions are Central and South America, Mediterranean Basin, South Africa, South Australia where the drought risk will increase p. 220
For plant species in the Mediterranean region, shifts in phenology, range contraction and health decline have been observed with precipitation decreases and temperature increases (medium confidence) (Settele et al., 2014). Recent studies using independent complementary approaches have shown that there is a regional-scale threshold in the Mediterranean region between 1.5°C and 2°C of warming (Guiot and Cramer, 2016; Schleussner et al., 2016b). Further, Guiot and Cramer (2016) concluded that biome shifts unprecedented in the last 10,000 years can only be avoided if global warming is constrained to 1.5°C (medium confidence) – whilst 2°C of warming will result in a decrease of 12–15% of the Mediterranean biome area.p 221
Generally, vulnerability to decreases in water and food availability is projected to be reduced at 1.5°C versus 2°C (Cheung et al., 2016a; Betts et al., 2018), especially in regions such as the African Sahel, the Mediterranean, central Europe, the Amazon, and western and southern Africa (medium confidence Box 6 238
Table 3.5 p. 247
Cross-Chapter Box 6 in this chapter highlights that at 2°C of warming, new literature shows that risks of food shortage are projected to emerge in the African Sahel, the Mediterranean, central Europe, the Amazon, and western and southern Africa, and that these are much larger than the corresponding risks at 1.5°C. p 256
IPCC WGI report
(…) projections show a warming of 0.6°C to 1.5°C, with highest changes over the land portion of the Mediterranean. Chapter 11, p. 991
Regional to global-scale projected decreases in soil moisture and increased risk of agricultural drought are likely in presently dry regions and are projected with medium confidence by the end of the 21st century under the RCP8.5 scenario. Prominent areas of projected decreases in evaporation include southern Africa and north western Africa along the Mediterranean. Soil moisture drying in the Mediterranean, southwest USA and southern African regions is consistent with projected changes in Hadley Circulation and increased surface temperatures, so surface drying in these regions as global temperatures increase is likely with high confidence by the end of this century underthe RCP8.5 scenario. Chapter 12 p1032
Significant increases in tropical nights are seen in southeastern North America, theMediterranean and central Asia. Chapter 12 p 1066
Areas with abundant atmospheric moisture availability and high present-day temperatures such as Mediterranean coastal regions are expected to experience the greatest heat stress changes because the heat stress response scales with humidity which thus becomes increasingly important to heat stress at higher temperatures Chapter 12 p 1066
Large increases in seasonal sea level pressure are also found in regions of sub-tropical drying such as the Mediterranean and northern Africa Chapter 12 p 1072
Decreased precipitation in the Mediterranean, Caribbean and Central America, southwestern United States and South Africa is likely under the RCP8.5 scenario and is projected with medium confidence to be larger than natural variations by the end of the 22nd century in some seasons
Chapter 12 p 1079
(…)consistency across the ensemble for drying in the Mediterranean region, northeast and southwest South America, southern Africa, and southwestern USA. Chapter 12 p 1079
The Mediterranean, southwestern USA, northeast South America and southern African drying regions are consistent with projected widening of the Hadley Circulation that shifts downwelling,
thus inhibiting precipitation in these regions. Chapter 12 p 1079
Evidence of climate risks to unique mountain ecosystems and their numerous endemic alpine species has continued to accrue in Europe, Asia, Australia, and South America (Sections 23.6.4, 24.4.2.3, 25.6.1, 27.3.2.1). Siberian, tropical, and desert ecosystems in Asia (Section 24.4.2.3), Africa (Warren et al., 2013a), and Mediterranean areas WGII Chapter 19 p 1075
The strongest warming of hot extremes is projected to occur in central and eastern North America, central and southern Europe, the Mediterranean region (including southern Europe, northern Africa and the Near East), western and central Asia, and southern Africa (medium confidence)
61-64. Long, awkward sentence. Suggest splitting into two sentences at “a significant part”.
28,256 ha
We have reworked the sentence
64-66. Consider consistently using the percent of total Spanish supply for both almonds and apples.
Here we are trying to show the relevance of both crops in Catalonia and Spain. In the case of almonds, what is relevant is the amount of almonds produced, not the percentage over Spanish production, which is world’s third largest. In the case of apples, Spanish production is not son relevant in the world, but Catalan apple production does in Spain’s context. That is the reason to present these figures in the way we do.
Don’t need both in-shell and without shell numbers, but be clear which you are giving.
We need to give both numbers as with shell is the number available in official statistics but without shell is the relevant one for producers.
65-66. Given the Catalonian almond supply is ~5% of Spanish total, seems irrelevant that Spain is 3rd in global almond supply.
What we intend here is to reflect the relevance of these crops either for Catalonia or for Spain. Then, we still believe that Spain being the third world almond producer is relevant
74-76. Bring up context of climate change as a driver for this research, but this research doesn’t put the CR found in context of future anticipated CRs. Consider Luedeling’s global CR projections for this.
We do not understand fully what the reviewer means by “future anticipated CRs” There is no indication that CR for a cultivar will change in the future. What is changing is the actual temperature and the difficulty to attain a certain amount of CR in a warmer climate. We just refer to the usefulness of CR and HR to anticipated that a cultivar will be fitted to a specific regions and/or a specific time in the future. However, we cannot address this issue unless we do not solve before the relationship between chill and heat accumulation phases, which we don’t in this manuscript.
106-108. Explain if/how Funes’s analysis was different from present analysis.
This information has been added to section 2.6, after introducing the method used here.
Are author’s concerns about using airport temperatures? Particularly the impact of radiative heat at night from paved surfaces influencing Tmin. Could seriously skew the data relative to temperatures experienced in the field.
We used the airport dataset because it was more complete than Mas de Bover AES, but we checked their equivalence with data of the common years. This information has been added to section 2.3
124, 128-129. Better to give p values in the results than list different levels of significance here.
We would generally agree with the Reviewer that giving p values would be better, but given the amount of data we prefer to keep, in this case the significance level scheme as we believe it makes easier to capture results in a glimpse.
135-138. That was not the point of Darbyshire or Campoy. Better to cite 19 & 20 again.
Done, thanks
138-140. This sentence can only be supported by papers that compared multiple ways of counting chill. 26 did not. This was not the point of Guo’s paper. Just cite 21.
We partially agree with the reviewer: although Guo papers does not compare different methods, it refers to the better performance of the Dynamic model. However, we agree that just keeping Benmoussa et al might be enough, although following considerations of reviewer 3, we also moved here references 3 and 34
150-151. Why cite Legave? Why not cite original PLS chill-heat papers?
“chilling and forcing” “established”
The reason to cite Legave is that it is one of the first papers to challenge sequentiallity. We have slightly modified the sentence to reflect it.
168-169. Unclear what statistical reasons are, what is meant by “independently”.
“until”
Here we refer to the fact that the criteria “first day of a period of consecutive days with persistent VIP values above 0.8 and negative standardized coefficients” were not met all along the phase: many days between the beginning and the end of each phase presented either VIP values below 0.8 or positive coefficients or both. Possible statistical reasons are discussed in section 4.1
The meaning of “independently” is explained in the next sentence. We have changed the stop to a colon to make it clearer.
“Till” changed to “until”
172-174. Does it concern researchers that forcing phase extended beyond mean bloom data in almost every case? Does this not seem to reveal a flaw/weakness in the PLS approach? If not, why not? Same re: 232-233.
This is discussed in section 4.1. lines 444 to 448 of the original manuscript
208-213. Why follow the classifications of Vargas and Romero when you have more than two times as much data as them, and your results contradict them? Why not make your own groupings of early-mid-late?
There are several reason for that: first, we do not agree that our results contradict theirs. In fact, as stated in section 3.1, “significant differences were found between all blooming groups except between late and extra-late bloomers”; as the extra-late group only includes two cultivars, we think that our results are mostly consistent with their classification. Second, we did not intend to classify the cultivars, so being our results generally consistent with a widely accepted classification, we did not feel a new classification was needed. To prepare a new classification would imply using classification multivariate techniques which not very often give unambiguous results, so frankly, we did not see the benefit of it.
Oct 23-24 is, roughly speaking, the average start found by the PLS analysis. That doesn’t mean it is truly when the trees/buds started counting each year.
The basis of PLS analysis is that if previous days were affecting blooming date, the year to year variation in chill accumulated in those days would have been detected by the analysis. In any case, from the comments of reviewer 2, we added an indication in sections 3.2.1, 3.2.2 and 4.1 that the amount of cold that could have been accumulated before that day would affect minimally CR.
235-238. What do the authors think is the significance of this lack of clear delination?
This is discussed in section 4.1
251-255. It is not clear how overlaps in counting or gaps in counting is shown in the figures.
Well, overlaps can be seeing as overlaps of pink and blue background colors (which identify chill and heat accumulation phases, respectively) and gaps as a white space between both background colors, as in ‘Jeromine’ in Fig 4. Anyway, it is better seen in Table 1.
161-262. But more overlap in mid-blooming when CR and HR was same as early blooming (e.g. Ramillete vs. Mollar, Rof, Alicante, Rana). This seems more interesting and somewhat new to me.
It can’t be otherwise given that the beginning of the chilling phase is common (although obtained independently for each cultivar): if the beginning is the same and CR and HR are the same, a different blooming date can only come from a different degree of overlapping (or gap).
What does “longed up” mean?
Prolonged. Changed
Table 1. This table is very hard to read with the wrapping text. Consider switching to landscape so all information for one cultivar is on one line.
“months” “established”
We will consider that in contact with the editor once the manuscript is accepted: switching to landscape implies spreading the table along several pages, which would also difficult reading it, and probably more pages for the paper.
Figure 4. Why aren’t VIP bars blue and green?
We believe blue bars for apples in the VIP panels (top part) help to differentiate from almonds, especially in the Supplementary Material, so we prefer to keep it.
No 3.4 Heading? Just skip to sub-heading?
Heading missing. Added.
Figure 5. Would be interesting to overlay bloom timing on this figure. Maybe color almond dots by earl/mid/late bloom.
We understand what the reviewer means, and it would work very well (early bloomers to the left, late bloomers to the right) were it not for overlaps and gaps. For instance, Ramillete and Garrigues would have the same color as Cavallera, Desmayo Largueta or Mollar de Tarragona, but because they have a higher HR and an overlap, they fall in the middle of the blue square in figure 5. At the end, the left square of figure 5 looks quite messy and the main difference is finally between left and right parts of the figure. We would prefer to keep the figure simple.
“X-axis, this implies” Use “Instead” rather than “On the contrary” “horizontal” To my eye, Cristomorto is the only cv. without a clear slope. If you’re going to differentiate, you need to give slope numbers, not just your visual assessment.
We agree that the value of these representations is limited: Slope numbers cannot be given, as these are not straight lines but curves: they just look roughly linear when figures are reduced as in figure 5 and 6. See, for instance, Ferraduel, Tuono, Prmiorsky… or please look at the individual figures in the supplemental material S35-S72.
371-376. Give summary statistics to support this assertion/differentiation.
377-383. Give summary statistics. Don’t just rely on visual assessment.
390-394. Need to put numbers to the slopes. Doesn’t look “nearly horizontal” to me.
No statistics available for figures 6 or 7: this is a graphical approach. See Guo and Luedeling papers.
402-404. Need regression statistics to support this. Chill phase end date vs. bloom date.
Supplemental table S9 show statistical differences in CR. As the initial date for chill accumulation is the same, it results in different ends for the chilling phase. The relation between blooming date and end of chilling phase (or CR) is only suggested. However, we have added a new figure 6 showing correlation (or lack or correlation) between blooming date (DOY), length of each accumulation phase, CR and HR, which support our statement.
23 or 22? I think you mean 22.
Yes
404-406. Do you really mean “On the other hand”? Isn’t this essentially the same = Late chill met à Late flowering, Early chill met à Early flowering
Yes, changed
409-420. How does this fit into the context of chill overlap work in almonds and apples of Pope et al (2014), Pope et al (2017) and Darbyshire et al (2017)?
This is mentioned in the following paragraph. There was an error in one of the references: reference 7 was cited instead of reference 9 (now references 9 and 11). Dealing with the mechanisms underlying overlapping or gapes between phases goes beyond our data. We have changed the sentence to a more assertive form and to exclude the lack of basis for gaps.
409-451. I see potential glimmers of how this work is novel in these paragraphs, but authors would need to make it more clear, and emphasis in the abstract, title and introduction.
We do not think the content of this section is strong enough to be reflected in the title. A mention has been added to the Introduction.
With this section in the Discussion, we just tried to present a possible limitation of the approach we used that would help to explain a relevant but not novel trend of our results, i.e., the presence of overlaps or gaps between the two phases. However, we believe PLS analysis as implemented in chillR is a valuable tool. We agree that all this should be reflected in the abstract, so we added a couple of final sentences in this respect, although this has resulted in exceeding the 200 words advised length
437-439. Is another possible explanation that the overlap is because the CR and HR are not consistently met on the same day in each year of the record?
We do not think so: the overlap does not appear in other statistical methods that have the same limitation when assuming the same day is the beginning or end of each phase along the years (e.g. Alonso).
This intrinsic to the method: using a common day for all years is ultimately false, but is only used to calculate an amount of chill and heat for each year. The relevant days are chosen based on the impact a variation in its temperature across years has in the recorded blooming days (VIP in PLS). This does not mean that the CP or GDH of that exact day (first or last day of each phase) are contributing each year to fulfill CR or HR, only that it did in average. In fact, blooming day for each year is also an average: a complete model should also explain why different flowers of the same tree open at different times if they experience the same temperatures.
Implicitly using an average implies a variance around it, but if this was the case, we would we expect having roughly 50% of gaps and 50% of overlaps, an our data are far from it.
458-531. Would be valuable, efficient and easier for the reader if you have a table of your estimates CRs and HRs, compared with the CR and HR estimates for the same cultivars from other works, e.g Benmoussa, Ramirez, El Yaccoubi. Rather than making the reader take your word for it that the numbers are similar or different, or taking up space listing the numbers, or making the reader go hunt down the numbers themselves to confirm what you are saying.
We are sorry, but we disagree with the reviewer: we already considered this possibility during the elaboration of the manuscript for several reasons:
The objective of the manuscript is not to give a final estimation of CR or HR. Unless the exact number is significant for the discussion, most papers confront their results with the literature as we did. This is not the case, as what is relevant here is if the method gives consistent results, not the precise results. The exact CR or HR has not a specific meaning demanding to go beyond the similarity or dissimilarity of values when comparing with the literature: for some parameters, being above or below a threshold has an implication and, hence, similar values from different studies would lead to different conclusions. This is not the case with CR or HR. Including all comparisons in a single table would mean repeating some results already presented in table 1, which is generally not accepted in scientific publications It would result in a heterogeneous table, with very few cultivars present, some of them being compared only HR (in some papers CR is not calculated with the Dynamic model, so it cannot be compared with our results), others with CR and HR from one, two or three papers depending on the precise cultivar.
Moreover, the reader does not need to take our word for those results (or not more that he/she will have to believe we have not invented all of our results): he or she can find them in the referenced papers and take his/her own judgement. And this is the usual procedure. He or she can choose between accepting we are honest all over the manuscript or check what has already been published. In fact, we do not agree with the idea of keeping a large part of our own work out of the manuscript (included in the Supplementary material) to include results that have already been published.
478-479. OK, different approaches yielded different results. But what numbers do you believe more and why? Given that El Yaacoubi was using a controlled experiment and PLS is just empirical, does this cast doubt on empirical findings? Is there a rational way to meld the two results together?
Awkward wording. Define “this”.
Yes, of course: we did not discuss this point as this is not the object of the paper, but the method used by El Yaacubi implies excising a part of a tree, so severing it from every relation with the rest of the plant (hormonal, for instance, but also microclimatic), and exposing it to forced climate conditions as opposite to natural conditions as used in the statistical approaches. So it makes their results as empirical and limited as ours.
Our point is not that our results are better or worse than each one of the particular papers we contrast them with, but that they generally agree with most of previous results using the same or a different method in the same or different areas, so we conclude they are widely consistent. The fact that we suggest some explanations in some cases where data do not match the literature does not mean we think other people’s results are wrong: in specific discrepancies, we cannot give a final conclusion on whether our results are better or worse than others, but as a whole we believe they are consistent and reliable. However, this might not evident in our original manuscript, so we have added a statement to make it clear.
We believe the Reviewer refers to “this” in line 491 (no other “this around, and specifically in lines 469-484): there, “this” refers to the opposed behavior of Primorsky in this manuscript and in Alonso. We have clarified that.
519-522. What is the physiological basis for including this “gap” in the model as a possibility? Do you think your results are more true to the real world experience of the trees/the true mechanism of chill-heat than Funes? And why?
Alonso or Funes? I think you mean Funes here.
The observed gap in this manuscript and other papers (e.g. Benmoussa) is a result of the statistical approach used, that does not link the beginning of the forcing phase to the end of the chill phase and estimates them independently. We do not know if this gap is real or a mere artifact of the method, and this has been discussed in section 4.1. Of course, if the gap can be proven true, a physiological basis will have to be provided for it, but this is beyond our reach at the moment. However, we took from Benmoussa et al one example of a possible physiological basis.
Funes uses Alonso’s approach, so we prefer to cite Alonso.
554-592. This section doesn’t integrate or analyze or contextualize any of the data from this paper. Either bringing in data and show what it means in the light of these predicted temperature increases, or delete whole section.
We have deleted the whole section as suggested here and by Reviewer 3.
715-717. Reformat all caps.
Done
Reviewer 2 Report
Diez et al. studied the chilling and heat requirements of 25 almond and 12 apple cultivars by using a statistical approach (Partial Least Square regression method) on long-term datasets. The period of interest consisted in 36 years for almond and 26 years for apple. The methods seem appropriate given the data and the discussion is sound. The results generated by this study may contribute to our understanding on dormancy requirements of almonds and apples in Mediterranean climate areas, such as northeast Spain and the basin of the Mediterranean sea. Below is my specific feedback.
Major issues
I think there is a different criterion for almond and apple when delineating the chilling phase (Figure 2 and 3). In both species, the beginning of the chilling phase is defined after a period of positive coefficient values. In almond however, this period seem to be preceded by a relatively stable sequence of negative coefficients (i.e. chill accumulation). For apples instead, a period of positive coefficients is considered within the chilling phase. Why use this different criterion?
Moreover, why for almonds cv. Garbí use as chilling phase between 24th Oct and 2nd Feb instead of between end of Sep to beginning of Jan? The period for negative coefficient values (i.e. chill accumulation) on Feb seem to be smaller compared to the positive values observed earlier.
Minor issues
L17-18: Consider adding Chill Portions and Growing Degree Hour in capital font. Moreover, add “growing” before “degree”.
L19: Consider to change “early and late bloomers” to “early and late blooming genotypes”.
L17-20: I would suggest splitting that sentence. Sometimes, it is difficult to follow very long sentences that include information between parentheses.
L17-20: Adding the number of years of records it may help.
L20-23: This sentence is not clear enough. After reading it several times, I think I got the point but I would suggest rephrasing. It could say: “The main results showed that the difference between early and late bloomers is due to chill requirements. Early bloomers had a CR of… “
L22-23: I would suggest to remove the sentence after comma. It can be highly expected that having 25 cultivars you get considerable variability in CR.
L23-25: As is now, this sentence contradicts the sentence above. Please clarify.
L26-27: I disagree the use of long sentences within parentheses. Under my point of view, those are important results from the study and should be placed in the main text.
L28: Consider including some final remarks, further implications or conclusions.
L29: Make it consistent between upper and lowercase font.
L36: Make it consistent between chilling and chill requirement. See abstract. Moreover, consider changing “dissected” to “the result”.
L38: I would propose that CR and HR give information of the species/cultivar but not of the region. Normally, growers try to match CR with cold conditions in a given region.
L43-45: I would suggest deleting “other” or change it to “some”.
L52-54: Consider modifying this sentence. As is now, it is a bit inaccurate since deciduous trees normally can resist very low temperatures during winter. Perhaps, it may be during early spring.
L57: Consider adding “Mediterranean climate areas”.
L61: Consider to add the author for the scientific name of apple as is now for almond.
L61-64: Consider specifying the relevance of the species by using percentages also. Sometimes, the number by itself does not says too much.
L64-66: Same as above.
L71: I would suggest using uppercase letters for “Chill Portions”, CP.
L73: Consider to delete “our case” and mention the specific name of the region of study.
L74: Add “n”.
L79: R Packages must be cited.
Figure 1: Consider deleting the red stars from the first zoom map. Moreover, make it clearer that triangle means weather station and star means A.E.S. Perhaps, in the legend.
Figure 1-Legend: Is it “Where” after (W.S)?
L104: A period is missing.
L130: Consider deleting this section. Chill and heat are not modeled but estimated only. Moreover, most of the readers should know about these models. Those are widely known and used. For more information, authors could visit the models references.
L146: Change “Chill R 0.62” to “chillR (version 0.62)”. Moreover, the citation (number 2) is not accurate. It seems more appropriate for citing the PLS method but not for the package. Besides that, in the reference list the reference for the package is version 0.70.2 please clarify.
L148: What is “It” referred to?
L147-151: I would suggest deleting this explanation since it is not accurate enough. The package does not assumes that chill and heat are continuously accumulated; instead, it contains some functions to compute chill and heat for a user-defined period.
L151: Change “Chill R” to “chillR”. Please verify through the whole document.
L154: Change “Chill R” to “PLS method” or any other denotation of the method.
L164: Consider changing “Seemingly” to “On the other hand”.
Figure 2: Change the orientation of the y-axes name (180°).
L197-199: This seems to belong to the manuscript template.
L223. Subheadings should have the same font size and format.
Figure 3. The figure looks blurry. Could the authors improve the quality?
Figure 3-Legend: Dynamic Model and GDH Model are both in uppercase. Please make it consistent throughout the whole document. Moreover, I would say that left and right panels mean the effect of chill and heat respectively. Chilling and heat phase are defined by the colors.
L245: in “…red bars” it seems there is an additional space, please check.
L291: Consider adding an “as” between “assumed” and “an”.
L293: Same as above for subheadings.
Table 1: Consider to modify the legend. As is now it is hard to understand the definitions for each parameter. I would suggest adding a first sentence to summarize the aim of the table and then continue to explaining the columns. Please make it clear when explaining one column and moving to the next one. Semicolon is not useful at all in this case. Please also check format for the first variety (bold and lines). In the legend, % CR and % Chill-Force overlap seem different columns. However, in the table % CR belong to % Chill-Force as a sub-category, please clarify. Please explain what it represents the number after the sign +/- in CR and HR columns. As is now, the column “% CR” is confusing since the sign (-) represents 100% of CR fulfilled. Consider including the actual value for clarity.
Continuation of table 1. If the columns mean the same as for almond’s table, I would suggest merging in only one table and including a new column or identification for the species name.
L342: The subheading is in the figure legend
L354-352: Make it consistent the use of either lower- or uppercase for chill and heat models name.
L402: Include the citation for “chillR”
L423: Consider to change “inadequacy” for “inaccuracy”
L413:415: It is not clear the relationship between this sentence and the exposed above. Please clarify.
L417: Consider to include “apple cv.” Before “Aporo”.
L440: “…reasons…”
L440-444: I would suggest authors to clearly discuss how those points (i, ii and iii) affect the analysis. Along the same lines, I agree there is a considerable autocorrelation between temperatures of consecutive days; however, the default running mean used in daily_chill function is 1. In that case, the statement “reinforced by the running mean used” is inaccurate. Please clarify, either here or in the methods, if a running mean was used and if so, how many days were taken.
L451: If the authors want to say that taking the big picture, PLS analysis is not that bad, I would suggest removing this sentence and giving a final section remark together with lines 455 – 457.
L459: Same as above for subheadings
L462-466: The difference for the varieties “Ferraduel” and “Tuono” between CR estimated by Benmoussa et al. (2017) and the authors is about 38 CP. In that case, I would suggest removing them from the list and keeping those ones with similar CR.
L466: Consider adding “… based on the use of PLS analysis…”
L469-472: I would suggest adding some values either in percentage of difference or absolute values.
L472-475: Is the single-node cutting method comparable in terms of response variable with authors’ methods? Did they measure the same phenological stage?
L474: “… the single-node cutting method …”
L488: change to “Benmoussa et al. [22]”. The reference 20 is for Fishman et al. (1987). Please verify all the citations are well referenced.
L508: “…compensated for each other…”
L555: “Based on the work…”
L559: Consider shifting “On the other hand,…” since it is normally used to say something opposite to the previous sentence.
References: Please make it consistent the use of abbreviations for journal names, capital letters and the font for scientific names.
Just a note: I am not a native English speaker but I would suggest revising the document to improve sentences like in L466, L508, L555, L559, and so on. Moreover, consider including more periods (.) instead of colons (:) for long sentences.

Author Response
Diez et al. studied the chilling and heat requirements of 25 almond and 12 apple cultivars by using a statistical approach (Partial Least Square regression method) on long-term datasets. The period of interest consisted in 36 years for almond and 26 years for apple. The methods seem appropriate given the data and the discussion is sound. The results generated by this study may contribute to our understanding on dormancy requirements of almonds and apples in Mediterranean climate areas, such as northeast Spain and the basin of the Mediterranean sea. However, under my point of view, the manuscript needs further revisions before it be published. Below is my specific feedback.
Major issues
I think there is a different criterion for almond and apple when delineating the chilling phase (Figure 2 and 3). In both species, the beginning of the chilling phase is defined after a period of positive coefficient values. In almond however, this period seem to be preceded by a relatively stable sequence of negative coefficients (i.e. chill accumulation). For apples instead, a period of positive coefficients is considered within the chilling phase. Why use this different criterion?
We believe the reviewer refers to figures 3 and 4. We do not agree we used different criteria: we do not take into account the positive coefficients but only a stable sequence of negative coefficients, as explained in mat and methods, section 2.6 . The difference comes from the different localization of their respective plots: for almonds, we discarded the small period of stable negative coefficients by end September- beginning October as there was almost no chill accumulated in that period. As this does not depend on cultivar but on climate, we decided to discard this period in all cases. We used the same criterion for apples, with chilling phase beginning somewhat later. We made it clearer in sections 3.2.1, 3.2.2 and 4.1 Finally, it would only affect minimally the CR as the chill accumulation in this period is very small.
Moreover, why for almonds cv. Garbí use as chilling phase between 24th Oct and 2nd Feb instead of between end of Sep to beginning of Jan? The period for negative coefficient values (i.e. chill accumulation) on Feb seem to be smaller compared to the positive values observed earlier.
The reason to discard end of September for almonds has just been explained. About the end, although the days in February have small coefficents in the PLS model, their relevance for the model is high, according to VIP value, and chill accumulation in those days is also high, so they cannot be discarded. The fact that some days in between show positive coefficients (alhough many are nonsignificant) is something we cannot explain, but this is a general difficulty in delineating all phases, recognized and discussed in section 4.1. We have consistently found this behavior in most late blooming almond cultivars.
Minor issues
L17-18: Consider adding Chill Portions and Growing Degree Hour in capital font. Moreover, add “growing” before “degree”.
Done
L19: Consider to change “early and late bloomers” to “early and late blooming genotypes”. We prefer to keep it shorter, and as the other two reviewers did not seem to be disturbed by this expression, we prefer to keep it. However, we changed it in different ways in the abstract.
L17-20: I would suggest splitting that sentence. Sometimes, it is difficult to follow very long sentences that include information between parentheses. We split it into three sentences, although this made the abstract longer.
L17-20: Adding the number of years of records it may help.
Done
L20-23: This sentence is not clear enough. After reading it several times, I think I got the point but I would suggest rephrasing. It could say: “The main results showed that the difference between early and late bloomers is due to chill requirements. Early bloomers had a CR of… “We split and shortened the sentence
L22-23: I would suggest to remove the sentence after comma. It can be highly expected that having 25 cultivars you get considerable variability in CR.
Done
L23-25: As is now, this sentence contradicts the sentence above. Please clarify. It does not: it might no be well expressed, but first sentence refers to intervarietal variability (variability among bloom date averages) and the second to intravarietal, year to year, variability. This is shown in sections 3.4.1 and 3.4.2 and discussed in section 4.3. We rephrased in an attempt to clarify this difference
L26-27: I disagree the use of long sentences within parentheses. Under my point of view, those are important results from the study and should be placed in the main text. Well, they are placed in the main text. We believe they are relevant enough to appear in the abstract as well. However, we have split the long sentence as we did for almonds.
L28: Consider including some final remarks, further implications or conclusions.
Done, although this has resulted in exceeding the 200 words advised length
L29: Make it consistent between upper and lowercase font.
Done
L36: Make it consistent between chilling and chill requirement. See abstract. Moreover, consider changing “dissected” to “the result”.
Done
L38: I would propose that CR and HR give information of the species/cultivar but not of the region. Normally, growers try to match CR with cold conditions in a given region.
Sentence has been deleted, except for the “cultivar-specific” nature of CR and HR, which has been moved to the end of the next sentence.
L43-45: I would suggest deleting “other” or change it to “some”.
We Agree, thanks.
L52-54: Consider modifying this sentence. As is now, it is a bit inaccurate since deciduous trees normally can resist very low temperatures during winter. Perhaps, it may be during early spring.
The sentence does not apply to any specific species, but is just a contrast to the detrimental effect of higher temperatures for some species: for others, it will be beneficial as present conditions are harmful, such as cold winters for citrus species above parallel 41 in the Iberian Peninsula. Areference to citrus species as an example has been introduced for clarity
L57: Consider adding “Mediterranean climate areas”.
It is true that Mediterranean climate areas are, in general, more vulnerable to climate change than other areas, but the Mediterranean is specifically vulnerable, according IPCC last reports.
L61: Consider to add the author for the scientific name of apple as is now for almond.
Done
L61-64: Consider specifying the relevance of the species by using percentages also. Sometimes, the number by itself does not says too much. L64-66: Same as above.
Done
L71: I would suggest using uppercase letters for “Chill Portions”, CP.
Yes. It was not uniform throughout the text. Same for Growing Degree Hours. Thank you.
L73: Consider to delete “our case” and mention the specific name of the region of study.
We prefer to keep the expression, because it encompasses both location and species, and for the location, we do not intend to generalize it to a specific region, as the locations we studied embrace coastal Catalonia, but they have performed well in Mediterranean climate areas and in warm regions.
L74: Add “n”.
Thanks
L79: R Packages must be cited.
It is in the appropriate section
Figure 1: Consider deleting the red stars from the first zoom map. Moreover, make it clearer that triangle means weather station and star means A.E.S. Perhaps, in the legend.
We prefer to keep the stars in the first map, but have added info to the figure caption
Figure 1-Legend: Is it “Where” after (W.S)?
Thanks
L104: A period is missing.
Thanks
L130: Consider deleting this section. Chill and heat are not modeled but estimated only. Moreover, most of the readers should know about these models. Those are widely known and used. For more information, authors could visit the models references.
We have change the section title to “Estimating”, but we prefer to keep this section: in previous manuscripts we had the same thought of the reviewer and were asked to include or enlarge the section.
L146: Change “Chill R 0.62” to “chillR (version 0.62)”. Moreover, the citation (number 2) is not accurate. It seems more appropriate for citing the PLS method but not for the package. Besides that, in the reference list the reference for the package is version 0.70.2 please clarify.
We disagree about citation number 2: in this paper, Luedeling et al make one of the first presentations of this package, which makes use of PLS package developed by Mevil et al.
About chillR version, it is continually evolving, and version 0.70.2 cited in ref 35 was simply released later: new versions do not change the calculation of CR or HR, they just give new tools to make calculations easier, present new graphs… Hence, here we just indicate the exact version we used
L148: What is “It” referred to?
“It” referred to the package, but we have substituted for clarity
L147-151: I would suggest deleting this explanation since it is not accurate enough. The package does not assumes that chill and heat are continuously accumulated; instead, it contains some functions to compute chill and heat for a user-defined period.
Our intention was to use this sentence as an introduction to the next sentence, i.e. that there is no assumption about the relationship between both phases. However, we agree that sentence in lines 148-151 is not necessary, so we have deleted it. But we have kept sentence of lines 147/148.which is equivalent to the explanation just given by the reviewer
L151: Change “Chill R” to “chillR”. Please verify through the whole document.
Verified
L154: Change “Chill R” to “PLS method” or any other denotation of the method.
It is true that dependent and independent variables refer to PLS, but we did not use PLS package directly but through chillR package, so we have introduced a reference to the PLS part of chillR.
L164: Consider changing “Seemingly” to “On the other hand”.
Done
Figure 2: Change the orientation of the y-axes name (180°).
Done
L197-199: This seems to belong to the manuscript template.
Ehem, true. Removed. Thank you.
L223. L293. L459. Subheadings should have the same font size and format.
Changed throughout
Figure 3. The figure looks blurry. Could the authors improve the quality?
Done. But this may be related to some compression process in uploading, downloading or sending the document: we have just re-inserted the same image from a tif file and now it looks ok to us.
Figure 3-Legend: Dynamic Model and GDH Model are both in uppercase. Please make it consistent throughout the whole document. Moreover, I would say that left and right panels mean the effect of chill and heat respectively. Chilling and heat phase are defined by the colors.
About uppercases, done. Left and right panels show how the chilling and the forcing phases were delineated, hence, the effect of chill and heat. The word was missing.
L245: in “…red bars” it seems there is an additional space, please check.
Done, thanks.
L291: Consider adding an “as” between “assumed” and “an”.
Yes, better.
Table 1: Consider to modify the legend. As is now it is hard to understand the definitions for each parameter. I would suggest adding a first sentence to summarize the aim of the table and then continue to explaining the columns. Please make it clear when explaining one column and moving to the next one. Semicolon is not useful at all in this case. Please also check format for the first variety (bold and lines). In the legend, % CR and % Chill-Force overlap seem different columns. However, in the table % CR belong to % Chill-Force as a sub-category, please clarify. Please explain what it represents the number after the sign +/- in CR and HR columns. As is now, the column “% CR” is confusing since the sign (-) represents 100% of CR fulfilled. Consider including the actual value for clarity.
We have reworked the caption deleting self-explained column titles (Start, End N. days of each phase). The % before Chill-Force overlap was meaningless: both %CR and % period are subcategories of Chill-Forcing, representing the %of CR fulfilled when the overlap begins, and the % of the chill period covered by the overlap, which ends with 100% CR fulfilled. WE hope it is clear now in the new caption. WE consider these % more informative than the actual value, as the relevance of the actual value is related to the total CR or the total duration of the chill period, respectively.
CR is the average chill accumulated in the years of the database from the during the chill phase of each year. The number after the ± sign is the standard deviation, as explained in the table caption. This is also explained in the last paragraph of section 2.6
Continuation of table 1. If the columns mean the same as for almond’s table, I would suggest merging in only one table and including a new column or identification for the species name.
We already considered that, but as the table will expand through two pages anyway, we preferred to show it as Continuation. Adding a new column just for the species name would narrow other columns and increase table size. In fact, reviewer 1 suggested just the contrary, to split it in two tables. We believe our intermediate solution is better.
L342: The subheading is in the figure legend
Not in our manuscript. We have checked in the revised manuscript to guarantee it is in the right place.
L354-352: Make it consistent the use of either lower- or uppercase for chill and heat models name.
Checked
L402: Include the citation for “chillR”
These are already in Materials and Methods, but ok.
L423: Consider to change “inadequacy” for “inaccuracy”
We prefer to keep “inadequacy”: what we would like to put on the table is not a problem of lack of accuracy (in the sense of being some days appart form the real day), but a lack of power of the method, despite PLS is designed to deal with this kind of situations. For instance, It is not unbelievable that some days have, just by chance, very similar temperatures in all or pretty most years in the database, so they cannot be detected as having an influence on bloom date. However, is the best statistical method, as far as we know.
L413:415: It is not clear the relationship between this sentence and the exposed above. Please clarify.
The sentence is not related to the previous part, but both ideas (interdependence of chill and heat phases and a blooming date somewhat forced by the breeding program) are related to the following sentence (tendency to overlap of long chill and heat phases…)
L417: Consider to include “apple cv.” Before “Aporo”.
Done
L440: “…reasons…”
Right, thanks.
L440-444: I would suggest authors to clearly discuss how those points (i, ii and iii) affect the analysis. Along the same lines, I agree there is a considerable autocorrelation between temperatures of consecutive days; however, the default running mean used in daily_chill function is 1. In that case, the statement “reinforced by the running mean used” is inaccurate. Please clarify, either here or in the methods, if a running mean was used and if so, how many days were taken.
We would be delighted to discuss it, but we have not data for that and it would be matter for another manuscript. We just offer some reasonable ideas of why PLS, which deals with a lot of uncertainty, may not be enough to deal with the interphase between chill and heat accumulation.
About the running mean, thank you, we had overlooked it in Materials and methods. We have added that information to M&M
L451: If the authors want to say that taking the big picture, PLS analysis is not that bad, I would suggest removing this sentence and giving a final section remark together with lines 455 – 457.
Done. We hope sentence added at the end of the section transmits this idea.
L462-466: The difference for the varieties “Ferraduel” and “Tuono” between CR estimated by Benmoussa et al. (2017) and the authors is about 38 CP. In that case, I would suggest removing them from the list and keeping those ones with similar CR.
The list refers to all common cultivars. The text already states that similar results are for most of the cultivars, not all, and that Ferraduel and Tuono have very different CR in both papers.
L466: Consider adding “… based on the use of PLS analysis…”
Done
L469-472: I would suggest adding some values either in percentage of difference or absolute values.
We do not think giving the precise values, either in percentage or absolute, would add anything to the reasoning, as the differences are already acknowledged.
L472-475: Is the single-node cutting method comparable in terms of response variable with authors’ methods? Did they measure the same phenological stage?
The main difference is that we used and statistical method based on data collected under natural conditions and the node-cutting method is an experimental method that uses excised tree parts under forced conditions. We have added a sentence later in the paragraph to clarify that.
L474: “… the single-node cutting method …”
Done
L488: change to “Benmoussa et al. [22]”. The reference 20 is for Fishman et al. (1987). Please verify all the citations are well referenced.
Thank you. We have rechecked the references several times, but this one escaped to us.
L508: “…compensated for each other…”
We found that there is some ambiguity in the use of this verb, but we accept the suggestion as it seems to fit better.
L555:L559:
This part has been removed following the comments of reviewers 1 and 3
References: Please make it consistent the use of abbreviations for journal names, capital letters and the font for scientific names.
Checked and corrected, thanks.
Just a note: I am not a native English speaker but I would suggest revising the document to improve sentences like in L466, L508, L555, L559, and so on. Moreover, consider including more periods (.) instead of colons (:) for long sentences.
We believe long sentences may be a problem of style, but not of language: long sentences do exist and are used in English texts written by native English speakers. Often, chopping interrelated ideas in shorter sentences obscures the interconnections to be shown or make the text much longer. Anyway, we have revised the document and rewritten some of them. Thanks for the note.
Reviewer 3 Report
The manuscript employs multiple types of analysis to infer responses from a great number of almond and apple cultivars. Overall the paper is well written, but could definitely be shortened – perhaps by aggregating the discussion that is currently separate for almonds and apples. At times it was a little overreaching, and some will be pointed out, but section 4.4 is an example: while it is an interesting aspect, there is no further analyses that justifies the whole section on the impact of climate change. That section could be greatly summarized into one paragraph.
Material and methods:
It should be indicated more clearly that the bloom dates used for part of the analyses were the average.
Results:
Table 1 should be split into table 1 for almonds and table 2 for apples.
I disagree with the authors in their assessment relevant to figure 5. The variation within the low chill almond group in heat requirement is similar to that of the apple cultivars. In fact, the variation in heat requirement is greater in almond than in apples.
Discussion:
The paragraph 421-439 is an example of how there is some overreach in the manuscript. The authors have to acknowledge that the results obtained are from modeling which is a probable cause of the differences from other references. In this paragraph the authors bring up gene expression too which may be exaggerating the conclusions from a model. In this section the authors might want to point out that other models for chill accumulation can be used where negation of chilling occurs (Utah and NC), but that for cold climates they’re very co-linear.
Specific comments below:
Line 33: I understand that frost damage is the commonly used term in Europe, but I would consider changing it to “cold damage during the winter” as to differentiate from freeze and frost damage during the spring.
Line 45: I think more references should be added for that specific affirmation considering the authors mention some species, but the reference is only in Douglas Fir. Some examples:
Couvillon and Erez, 1985 – multiple fruit species
Citadin et al., 2001 – peach
Hunter and Lechowicz, 1992 – 26 North American native species
Okie and Blackburn, 2011 - peach
Dantec et al., 2014 – European beech and sessile oak
Kovaleski et al., 2018 – grapevines
Line 53: remove “e.g. apple and almond”
Lines 54-56: That sentence is just a repetition of previous. Could be removed.
Line 59: Reference 14 is the 2014 IPCC report. I’m not fully up to date in this, but does the new 2018 report have similar information? Just to make this more current.
Lines 64-67: Use either % contribution to total of Spain or weight (Almonds is in tons and apple is % production).
Line 69: I don’t think the Dynamic model is the most frequently used chill model. Luedeling’s papers have mentioned that some people refrain from using it because it is not easy to compute (or wasn’t until his chillR package). It is the one with the most translatable results for different regions though, so perhaps some changing of the language is required here.
Lines 71-72: Separate the abbreviations: “Both attribute a specific parameter to quantify chill (chill portions, CPs) and heat effects (GDH).”
Line 106: “from 1992 to 2018”
Line 109: The reference is for the SAS licence, not ANOVA. Change to: “through ANOVA using SAS [31]”
Lines 135 and 138: The references there should really be the Fishman et al. papers [19,20], as those sentences are referring to how the model is built/behaves. The other references are fitting of the end of the paragraph (Line 140).
Line 192: Change were to where.
Lines 197-199: Remove.
Lines 201-204: This is a one sentence paragraph. I suggest bringing up the following paragraph (another one sentence), and rewriting 201-204 as follows:
“Temperature trends were only significant for average and maximum, but not minimum. Annual temperature rose at a decadal rate of …”
Line 224: Change “on the late-October” to “in late October”
Line 225: 23-24 October (no “th”) – remove it from other dates as well (E.g. line 229, table 1).
Line 230: Change “till” (informal) to “until”
Line 318: Change “moths” to “months”
Lines 349-351: Should be in figure legend.
Lines 428-429: remove the “for” before pistachio, walnut, cherry, chestnut and jujube, and apricot.
Line 451: Not necessary to mention whether the results soften the criticism or not.
Line 467: Not sure what the authors mean by “different people”. Is it different observers for phenological measurements?
Lines 478-479: See above about describing other forms of chill accumulation.
Line 501: abbreviation for coefficients of variation is introduced and is not used again. Since it is only used twice, should perhaps refrain from introducing another abbreviation.
Line 528-529: This should be removed. There is interest in different times of blooming for a plethora of reasons in fruit crops, including apple.
Lines 533-536: Great sentence. Perhaps a good introduction to mentioning effects of climate change while removing section 4.4.
Lines 610-613: one sentence paragraph.
Figure 2: Text says there is no difference between late blooming and extra-late blooming, so I’m not sure why they are actually separated in the figure. The supplementary info is not available, so I can’t assess that part.
Figures 3 and 4:
The overlapping of chill and heat phase are very subtle when printed. Perhaps adding texture could help. I think colors should be switched: green should be the contributing accumulation and red the non-contributing. Figure 4 specifically has blue VIP bars. The panels should probably include a cumulative chill and heat figure to demonstrate how much it goes up over time (the integration of the bottom graph).
Author Response
The manuscript employs multiple types of analysis to infer responses from a great number of almond and apple cultivars. Overall the paper is well written, but could definitely be shortened – perhaps by aggregating the discussion that is currently separate for almonds and apples. At times it was a little overreaching, and some will be pointed out, but section 4.4 is an example: while it is an interesting aspect, there is no further analyses that justifies the whole section on the impact of climate change. That section could be greatly summarized into one paragraph.
The section has been removed after Reviewer 1 and Reviewer 3 comments
Material and methods:
It should be indicated more clearly that the bloom dates used for part of the analyses were the average.
Does the reviewer mean for the end of the forcing phase, for those cultivars that presented days with high VIP after the mean blooming date? If so, this is already stated in section 2.6 and discussed in section 4.1
If the reviewer refers to other uses of average blooming dates, such as % of chill-force overlap or gap in Table 1, it derives from the former. In all other cases (PLS, past climatic or blooming trends, variance analyses) all data were used, not averages.
Results:
Table 1 should be split into table 1 for almonds and table 2 for apples.
We prefer to keep it as a single table, as the only difference is the species and changing to table 2 will not save any space or make the table more readable. In fact, reviewer 2 suggested just the contrary, to join both table in a single table. We believe our intermediate solution is better.
I disagree with the authors in their assessment relevant to figure 5. The variation within the low chill almond group in heat requirement is similar to that of the apple cultivars. In fact, the variation in heat requirement is greater in almond than in apples.
We do not understand exactly to which part of the assessment the reviewer refers: it is true that the low chill almond group has a similar variation in HR but, first, it is not the same range, and second, we compared all almond cultivars, not a part of them. If all almond cultivars are taken, it is clear that both CR and HR present a higher variability than apple cultivars.
Moreover, what we affirm about figure 5 is that clear groups can be formed in almonds and not in apples, and that almond groups are separated by CR, which is clear in the figure. The last sentence in section 3.3.2 might be confusing as it refers not only to figure 5 but also to bloom dates, which are in Table 1. A reference to Table 1 has been added
Discussion:
The paragraph 421-439 is an example of how there is some overreach in the manuscript. The authors have to acknowledge that the results obtained are from modeling which is a probable cause of the differences from other references. In this paragraph the authors bring up gene expression too which may be exaggerating the conclusions from a model. In this section the authors might want to point out that other models for chill accumulation can be used where negation of chilling occurs (Utah and NC), but that for cold climates they’re very co-linear.
We do not agree: all the comparisons in this paragraph are with papers using the same approach, and are used to show that the appearance of overlaps or gaps is not exclusive of our results. Hence, we think that a reference to other chill accumulation models is not pèrtinent here.
The reference to gene expression is taken from another paper using the same method and is an effort to give some phisiological basis to gaps, provided they are proven true. In fact, reviewer 1 asked for more sound physiological basis for gaps. Moreover, we state, where pertinent, that the difference in results with the literature in some cultivars arises from different methodologies being used (statistical vs experimental: see section 4.2.1).
Specific comments below:
Line 33: I understand that frost damage is the commonly used term in Europe, but I would consider changing it to “cold damage during the winter” as to differentiate from freeze and frost damage during the spring.
Done
Line 45: I think more references should be added for that specific affirmation considering the authors mention some species, but the reference is only in Douglas Fir. Some examples:
Couvillon and Erez, 1985 – multiple fruit species
Citadin et al., 2001 – peach
Hunter and Lechowicz, 1992 – 26 North American native species
Okie and Blackburn, 2011 - peach
Dantec et al., 2014 – European beech and sessile oak
Kovaleski et al., 2018 – grapevines
We appreciate the references offerd by the reviewer. We do not think we need to be exhaustive here, so we added two of them
Line 53: remove “e.g. apple and almond”
Done
Lines 54-56: That sentence is just a repetition of previous. Could be removed.
We do not think they mean exactly the same, but the order of the sentences may obscure it: we have reordered the last part of the paragraph to make it clearer.
Line 59: Reference 14 is the 2014 IPCC report. I’m not fully up to date in this, but does the new 2018 report have similar information? Just to make this more current.
Done, updated to the Special Report on Global Warming of 1.5 ⁰C: although the perspective is not the same, the bottom line does.
Lines 64-67: Use either % contribution to total of Spain or weight (Almonds is in tons and apple is % production).
Here we are trying to show the relevance of both crops in Catalonia and Spain. In the case of almonds, what is relevant is the amount of almonds produced, not the percentage over Spanish production, which is world’s third largest. In the case of apples, Spanish production is not son relevant in the world, but Catalan apple production does in Spain’s context. That’s the reason to present these figures in the way we do.
Line 69: I don’t think the Dynamic model is the most frequently used chill model. Luedeling’s papers have mentioned that some people refrain from using it because it is not easy to compute (or wasn’t until his chillR package). It is the one with the most translatable results for different regions though, so perhaps some changing of the language is required here.
That’s right: Dynamic Model is one of the most used in scientific papers in warm areas, no in general or in technical information. We have changed the sentence to reflect that.
Lines 71-72: Separate the abbreviations: “Both attribute a specific parameter to quantify chill (chill portions, CPs) and heat effects (GDH).”
Done, thanks
Line 106: “from 1992 to 2018”
Done, thanks
Line 109: The reference is for the SAS licence, not ANOVA. Change to: “through ANOVA using SAS [31]”
Reformulated
Lines 135 and 138: The references there should really be the Fishman et al. papers [19,20], as those sentences are referring to how the model is built/behaves. The other references are fitting of the end of the paragraph (Line 140).
Changed, thank you. However, following reviewer 1 comments, we have deleted references to Guo papers from here (formerly refernces 25, 26, now 27-28.
Line 192: Change were to where.
Done
Lines 197-199: Remove.
Done
Lines 201-204: This is a one sentence paragraph. I suggest bringing up the following paragraph (another one sentence), and rewriting 201-204 as follows:
“Temperature trends were only significant for average and maximum, but not minimum. Annual temperature rose at a decadal rate of …”
Done, thanks.
Line 224: Change “on the late-October” to “in late October”
Thanks
Line 225: 23-24 October (no “th”) – remove it from other dates as well (E.g. line 229, table 1).
We followed here “MDPI English editing guidelines for authors”, which is to be applied to this special Issue. It states: “The 'th' in 19th or 20th should NOT be written in superscript.” Hence, we assume then the “th” is supposed to appear.
Line 230: Change “till” (informal) to “until”
Done
Line 318: Change “moths” to “months”
Done
Lines 349-351: Should be in figure legend.
Information about isolines has been added to the figure legend.
Lines 428-429: remove the “for” before pistachio, walnut, cherry, chestnut and jujube, and apricot.
Done
Line 451: Not necessary to mention whether the results soften the criticism or not.
Following comments of Reviewer 2, the sentence has been rewrtitten and moved to the end of the section
Line 467: Not sure what the authors mean by “different people”. Is it different observers for phenological measurements?
We mean here not the observers but authors of different papers in the literatures using chillR to delineate phenological phases and, then, including subjectivity in the process. We have change “people” to “authors and chillR users” to clarify it.
Lines 478-479: See above about describing other forms of chill accumulation.
Please see our answer there.
Line 501: abbreviation for coefficients of variation is introduced and is not used again. Since it is only used twice, should perhaps refrain from introducing another abbreviation.
Right. The abbreviation is also used in Table 1, but it is defined there. Removed.
Line 528-529: This should be removed. There is interest in different times of blooming for a plethora of reasons in fruit crops, including apple.
We referred to the specific set of apple cultivars used here as, in contrast with the specific set of almond cultivars, do not share a common interest in flowering date in the respective breeding programs. Rewritten to clarify it.
Lines 533-536: Great sentence. Perhaps a good introduction to mentioning effects of climate change while removing section 4.4.
Section 4.4 has been removing, following also comments of Reviewer 1.
Lines 610-613: one sentence paragraph.
Yes, but we prefer to keep it as it is, as this is a kind of final paragraph to close the conclusion section and is not more related to the previous paragraphs than to all other paragraphs in this section. Moreover, we have split it in two sentences.
Figure 2: Text says there is no difference between late blooming and extra-late blooming, so I’m not sure why they are actually separated in the figure. The supplementary info is not available, so I can’t assess that part.
As stated in the figure legend, almond cultivars are organized in the figure following the classification of Vargas and Romero [40]. What can be seen in the figure is that our results fit into their classification, except for late and extra-late, than would correspond to a single group according to our results.
We are sorry to learn the Supplementary Information has not been made available to the Reviewers. It was sent together with all other parts.
Figures 3 and 4:
The overlapping of chill and heat phase are very subtle when printed. Perhaps adding texture could help. I think colors should be switched: green should be the contributing accumulation and red the non-contributing. Figure 4 specifically has blue VIP bars. The panels should probably include a cumulative chill and heat figure to demonstrate how much it goes up over time (the integration of the bottom graph).
We are aware of the difficulties to clearly see overlapping under some conditions, we tried other combination of colors and could not improve the results (less transparent colors obscured the grey bars (i.e. non-green and non-red).
About bar colors, we have followed previous conventions for chillR results and prefer to keep it this way.
We believe blue bars for apples in the VIP panels (top part) help to differentiate from almonds, especially in the Supplementary Material, so we prefer to keep it.
The figure suggested by the Reviewer might interesting and informative. However, we prefer not to include it to avoid adding somewhat redundant information in an already overcrowded manuscript.
